# Riemannian Consistency Model

**Chaoran Cheng**[*1], **Yusong Wang**[*2], **Yuxin Chen**[1], **Xiangxin Zhou**[3], **Nanning Zheng**[2], **Ge Liu**[1]
[1]University of Illinois Urbana-Champaign,
[2]Xi'an Jiaotong University, [3]University of Chinese Academy of Sciences

## Abstract

Consistency models are a class of generative models that enable few-step generation for diffusion and flow matching models. While consistency models have achieved promising results on Euclidean domains like images, their applications to Riemannian manifolds remain challenging due to the curved geometry. In this work, we propose the Riemannian Consistency Model (RCM), which, for the first time, enables few-step consistency modeling while respecting the intrinsic manifold constraint imposed by the Riemannian geometry. Leveraging the covariant derivative and exponential-map-based parameterization, we derive the closed-form solutions for both discrete- and continuous-time training objectives for RCM. We then demonstrate theoretical equivalence between the two variants of RCM: Riemannian consistency distillation (RCD) that relies on a teacher model to approximate the marginal vector field, and Riemannian consistency training (RCT) that utilizes the conditional vector field for training. We further propose a simplified training objective that eliminates the need for the complicated differential calculation. Finally, we provide a unique kinematics perspective for interpreting the RCM objective, offering new theoretical angles. Through extensive experiments, we manifest the superior generative quality of RCM in few-step generation on various non-Euclidean manifolds, including flat-tori, spheres, and the 3D rotation group SO(3), spanning a variety of crucial real-world applications such as RNA and protein generation.

## 1 Introduction

Diffusion [47, 48, 19] and flow matching [27, 7] models have achieved remarkable success on generative modeling in various domains including image generation [42, 11], protein design [55, 3], and text generation [2, 14, 9]. As an intrinsically iterative process that gradually transforms the data from random noises into meaningful samples, the inference procedure of diffusion and flow matching usually requires hundreds to up to a thousand steps for decent generation. To mitigate such high computational cost, a new family of generative models known as the Consistency Model (CM) [50] was proposed. By "shortcutting" the probability flow and enforcing consistent model outputs, consistency models are able to generate high-quality samples using one or a few steps. In the image generation task, consistency models have surpassed the existing distillation approaches [44] and rectified flows [29, 28] in one-step generation.

Besides Euclidean domains like images, generative models on various Riemannian manifolds have a potentially profound impact in scientific domains, including protein generation [55, 3], peptide design [26, 25], robotics [5], and geoscience [41]. For example, the generative modeling of a protein requires descriptors of its position, orientation, and torsion angles for each amino acid. While the position is Euclidean, the orientation lies in the 3D rotation group SO(3) and the torsion angle lies in

---

[*]Equal contribution.
[†]Corresponding author to `chaoran7@illinois.edu`.

the flat torus where a translation by $2\pi$ will result in the same angle. Existing works for protein design typical require 200 to 1000 sampling steps, limiting the overall throughput for virtual screening. A fast and effective few-step generative model on Riemannian manifolds, therefore, can significantly accelerate the drug discovery and enzyme design processes, further facilitating crucial real-world pharmaceutical applications.

While the current Euclidean consistency model has achieved superior performance on images where the distance between two predictions is measured with the standard Euclidean norm, its counterpart in Riemannian manifolds, the **Riemannian consistency model (RCM)**, poses additional challenges and remains largely unexplored. Specifically, the intrinsically curved manifold requires the consistency parameterizations to lie on the manifold. In this way, simple linear interpolation like Lu and Song [33], Yang et al. [54] is not always feasible. Furthermore, such a manifold constraint will impose an additional constraint on the vector field when the consistency loss is enforced, that is, the vector field at different points should also lie in their corresponding tangent spaces, necessitating additional corrections in the time derivative.

In this work, we address the challenge of Riemannian Consistency Modeling with a novel consistency parameterization based on the exponential map to ensure the manifold constraint and use the *covariant derivative* to account for the intrinsic curved geometry when calculating the time derivative. We provide the closed-form solutions for both discrete- and continuous-time formulations of the RCM objective. Similar to the Euclidean CM, we theoretically prove that the Riemannian consistency distillation (RCD), which relies on a teacher model to approximate the marginal vector field, can be extended to Riemannian consistency training (RCT), which directly utilizes the conditional vector field with marginalization techniques for training. We further propose a simplified training objective that empirically improves model performance and eliminates the need for potentially complicated calculations of the differentials of the exponential map. Intriguingly, we provide an intuitive interpretation of the RCM objective from the perspective of kinematics on curved geometries, offering new theoretical angles. We carry out extensive experiments on non-Euclidean manifolds, including flat tori, spheres, and the 3D rotation group $SO(3)$. Compared to the vanilla flow-matching and the naive Euclidean adaptation of the consistency model, our RCM recipe demonstrates higher-quality generations with better distributional fitness in the few-step generation setting.

## 2 Preliminary

### 2.1 Riemannian Flow Matching

Conditional flow matching (CFM) [27] learns a time-dependent vector field that pushes the prior noise distribution to any target data distribution. Such a flow-based model can be viewed as the continuous generalization of the score matching (diffusion) model [47, 48, 19] while allowing for a more flexible design of the denoising process. Riemannian flow matching (RFM) [7] further extends CFM to general manifolds on which a well-defined distance metric can be computed.

Mathematically, consider a smooth Riemannian manifold $\mathcal{M}$ with the Riemannian metric $g$, a *probability path* $p_t : [0, 1] \to \mathcal{P}(\mathcal{M})$ is a curve of probability densities over $\mathcal{M}$. A *flow* $\psi_t : [0, 1] \times \mathcal{M} \to \mathcal{M}$ is a time-dependent diffeomorphism defined by a time-dependent vector field $u_t : [0, 1] \times \mathcal{M} \to T\mathcal{M}$ via the probability flow ordinary differential equation (PF-ODE): $\frac{\mathrm{d}}{\mathrm{d}t}\psi_t(x) = u_t(\psi_t(x))$. The flow matching objective directly regresses the conditional vector field $u_t(x_t|x_0, x_1) := \frac{\mathrm{d}}{\mathrm{d}t}x_t$ with a time-dependent neural net $v_\theta(x_t, t)$ where $x_t := \psi_t(x)$. The Riemannian flow matching objective can be formulated as:

$$\mathcal{L}_{\text{RFM}} = \mathbb{E}_{t \sim U[0,1], x_0 \sim p_0(x), x_1 \sim q(x)} \left[ \|v_\theta(x_t, t) - u_t(x_t|x_0, x_1)\|_g^2 \right], \tag{1}$$

where $q$ is the data distribution, $p_0$ is the prior distribution, and $x_t := \psi_t(x|x_0, x_1)$ denotes the conditional flow. Chen and Lipman [7] further demonstrated that if the exponential map and logarithm map can be evaluated in closed-form, the condition flow can be defined as the geodesic interpolation $x_t = \exp_{x_1}(\kappa_t \log_{x_1} x_0)$, where $\kappa_t$ is a monotonically decreasing schedule satisfying $\kappa_0 = 1, \kappa_1 = 0$. In this way, the corresponding vector field can be calculated as $u_t(x_t|x_0, x_1) = \dot{\kappa}_t \log_{x_t} x_1 / \kappa_t$. In this work, we use $\dot{x}, \dot{v}$, etc, to denote the time derivative of $x, v$ and follow the Einstein summation notation. If necessary, the time $t$ will be noted in the additional subscript.

## 2.2 Consistency Model

The marginal vector field learned by the (Euclidean) flow matching model is not necessarily straight, which requires solving the PF-ODE with hundreds of iterative steps. The consistency model (CM) [50], by design, can generate high-quality samples by directly mapping noise to data. CM achieves one-few generation by "short-cutting" the PF-ODE such that the denoiser $f_\theta(x_t, t)$ along the PF-ODE should output consistency predictions:

$$\mathcal{L}_{\text{CM}}^N = N^2 \mathbb{E}_{t, x_t} \left[ w_t \| f_\theta(x_t, t) - f_{\theta^-}(x_{t+\Delta t}, t + \Delta t) \|_2^2 \right], \tag{2}$$

where $\Delta t = 1/N$ is the discretization step, $\theta^-$ is the stop-gradient operation, and $w_t$ is a weighing function. $x_{t+\Delta t}$ is defined by following the marginal vector field at $x_t$ along the PF-ODE. To avoid trivial solutions, an additional consistency constraint $f_\theta(x_1, 1) = x_1$ needs to be enforced. Song et al. [50] utilized a parameterization of $f_\theta(x_t, t) := c_{\text{in}}(t)x_t + c_{\text{out}}(t)F_\theta(x_t, t)$ where $F_\theta(x_t, t)$ is the unconstrained denoiser and $c_{\text{in}}(t), c_{\text{out}}(t)$ are schedulers such that $c_{\text{in}}(1) = 1, c_{\text{out}}(1) = 0$. Yang et al. [54], Lu and Song [33] further related the denoiser parameterization with the vector field (score) parameterization. In practice, a pre-trained flow matching model is used to approximate the marginal vector field (consistency distillation, CD). Additionally, Song et al. [50] also demonstrated that, with the stop-gradient operation, training on the conditional vector field (consistency training, CT) has the same marginalization effect. Existing CMs have achieved high-quality one-step or few-step generations on Euclidean domains like image generation. Their Riemannian counterpart, however, remains unexplored.

# 3 Riemannian Consistency Model

## 3.1 Consistency Model on Riemannian Manifolds

As discussed above, the extension of the consistency model to Riemannian manifolds poses additional challenges, as the model needs to be fully aware of the intrinsic geometry to shortcut the Riemannian PF-ODE. To address the above challenges, we choose to learn the vector field $v_\theta(x_t, t)$ and adopt the consistency parameterization utilizing the exponential map to ensure the manifold geometry (see Figure 1):

$$f_\theta(x_t, t) := \exp_{x_t} \kappa_t v_\theta(x_t, t). \tag{3}$$

It is easy to verify that for any bounded vector field $v_\theta(x_t, t)$, the above parameterization satisfies the consistency constraint $f_\theta(x_1, 1) = x_1$ as $\kappa_1 = 0$.

The geometric property of Riemannian manifolds allows us to define *geodesics* as the "straight lines"

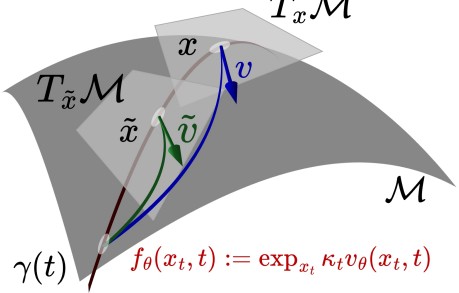

Figure 1: Riemannian Consistency Model (RCM). The denoiser $f_\theta$ along the Riemannian PF-ODE should be consistent.

on the manifold, whose lengths are *geodesic distance*, the shortest distance between two points on the manifold. Therefore, following the core idea that the predictions along the same PF-ODE should be consistent, we use the geodesic distance to measure the consistency:

$$\mathcal{L}_{\text{RCM}}^N = N^2 \mathbb{E}_{t, x_t} \left[ w_t d_g^2 \left( f_\theta(x_t, t), f_{\theta^-}(x_{t+\Delta t}, t + \Delta t) \right) \right]. \tag{4}$$

Similar to Euclidean cases, the next data point $x_{t+\Delta t}$ is defined along the marginal PF-ODE on the manifold. When $N \to \infty$, we have the following continuous-time limit of RCM:

**Theorem 3.1.** *When $N \to \infty, \Delta t \to 0$, the continuous-time RCM loss is*

$$\mathcal{L}_{\text{RCM}}^\infty := \lim_{N \to \infty} \mathcal{L}_{\text{RCM}}^N = \mathbb{E}_{t, x_t} \left[ w \left\| d(\exp_x)_u \left( \dot\kappa v + \kappa \nabla_{\dot x} v \right) + d(\exp u)_x \left( \dot x \right) \right\|_g^2 \right], \tag{5}$$

*where $d(\exp_x)_u, d(\exp u)_x : T_x\mathcal{M} \to T_{f(x)}\mathcal{M}$ are the differentials of the exponential map with respect to the tangent vector $u = \kappa v$ and the base point $x$. $\nabla_{\dot x}$ denotes the covariant derivative with respect to the Levi-Civita connection of the Riemannian manifold $(\mathcal{M}, g)$ along the PF-ODE.*

*Proof.* Note that $d_g^2(x, y) = \| \log_x y \|_g^2$ holds for any $x, y \in \mathcal{M}$. As $\Delta t \to 0$, the two target points in Eq.4 also approach each other. Therefore, the first-order approximation $\log_x y \approx y - x$ holds. Now consider $(f_\theta(x_{t+\Delta t}, t + \Delta t) - f_\theta(x_t, t))/\Delta t \to \dot{f}$. Applying the chain rule, we get:

$$\dot{f} = \mathrm{d}(\exp_x)_u \left( \dot{\kappa} v + \kappa \nabla_{\dot{x}} v \right) + \mathrm{d}(\exp u)_x \left( \dot{x} \right). \tag{6}$$

Marginalization over $t$ and $x_t$ concludes the proof. $\qquad\square$

In the above formulation, we assume that the marginal vector field $\dot{x}$ is available via the pre-trained RFM model, leading to the **Riemannian consistency distillation (RCD)** where the marginal $\dot{x}$ is learned by a pre-trained Riemannian flow matching model. Similar to [50], in the following theorem, we demonstrate that the same objective can be utilized with the conditional vector field $\dot{x}|x_1$, leading to **Riemannian consistency training (RCT)**.

**Theorem 3.2.** *With the stop-gradient operator $\theta^-$ in Eq.4, the marginal vector field $\dot{x}$ in the RCD loss can be substituted with the conditional vector field $\dot{x}|x_1$.*

The following lemma is the key result for extending RCD to RCT:

**Lemma 3.1.** $\dot{f}$ *is linear in* $\dot{x}$.

*Proof.* By the definition of covariant derivative, $\nabla_{\dot{x}}$ is linear with respect to $\dot{x}$. Also note that $\mathrm{d}(\exp_x)_u, \mathrm{d}(\exp u)_x$ are both linear mappings. Therefore, the final result is linear in $\dot{x}$. $\qquad\square$

**Lemma 3.2.** $\dot{x} = \mathbb{E}[(\dot{x}|x_1)|x_t]$, *or using the Riemannian integral* $\int_{\mathcal{M}} u(x|x_1) p_t(x_1|x) \, \mathrm{dvol}_{x_1}$.

Lemma 3.2 generalizes the Euclidean case and can be verified using the Bayes' rule on Riemannian manifolds [6]. It draws the connection between the conditional and marginal vector fields. The key to the proof of Theorem 3.2 is that we want to move the expectation outside to use the conditional vector fields.

*Proof for Theorem 3.2.* Let $\tilde{f}$ denote the denoising result in Eq.3 at time step $t + \Delta t$ and $\Delta f = \tilde{f} - f$. Note that $d_g^2(f_\theta, \tilde{f}_\theta) = \| \log_{f_\theta} \tilde{f}_\theta \|_g^2$. Taking the gradient on both sides, we have

$$\begin{aligned} \frac{1}{2} \nabla_\theta d_g^2(f_\theta, \tilde{f}_{\theta^-}) &= \frac{1}{2} \nabla_\theta \langle \log_{f_\theta} \tilde{f}_{\theta^-}, \log_{f_\theta} \tilde{f}_{\theta^-} \rangle_g = \langle \log_{f_\theta} \tilde{f}_{\theta^-}, \nabla_\theta \log_{f_\theta} \tilde{f}_{\theta^-} \rangle_g \\ &\approx \langle \Delta f_\theta, \nabla_\theta (\tilde{f}_{\theta^-} - f_\theta) \rangle_g = -\langle \Delta f_\theta, \nabla_\theta f_\theta \rangle_g. \end{aligned} \tag{7}$$

Using Lemma 3.1, $\frac{\Delta f_\theta}{\Delta t} \to \dot{f}$ is linear in $\dot{x}$ and the second argument $\nabla_\theta f_\theta$ is now independent of $\dot{x}$. This indicates $\nabla_\theta d_g^2(f_\theta, \tilde{f}_{\theta^-})$ is also linear in $\dot{x}$. Therefore, when using Lemma 3.2 to marginalize over $\dot{x}$, we can simply move the expectation over $x_t$ outside the gradient operation, leaving the conditional vector field $\dot{x}|x_1$ inside the expectation. $\qquad\square$

The proof above inspires us to use an alternative loss as

$$\mathcal{L}_{\text{RCM}}^\infty := \mathbb{E}_{t,x_t} \left[ w \left\langle f_{\theta^-} - f_\theta + \dot{f}_{\theta^-}, \dot{f}_{\theta^-} \right\rangle_g \right], \quad \dot{f} = \mathrm{d}(\exp_x)_u \left( \dot{\kappa} v + \kappa \nabla_{\dot{x}} v \right) + \mathrm{d}(\exp u)_x \left( \dot{x} \right) \tag{8}$$

where the differentials are calculated along conditional vector fields. It is easy to verify that the loss in Eq.8 has the same value as Eq.5 and the same gradient as demonstrated in the proof above.

## 3.2 Simplified Riemannian Consistency Model

The loss in Eq.8 involves the calculation of the differentials of the exponential map $\mathrm{d}(\exp_x)_u, \mathrm{d}(\exp u)_x$, which may require additional complex symbolic calculation for efficient implementation (see Appendix B). Instead, we proposed an alternative loss on the vector fields that eliminates the need to compute these differentials:

$$\mathcal{L}_{\text{sRCM}}^\infty := \mathbb{E}_{t,x_t} \left[ w \| \dot{x} + \dot{\kappa} v + \kappa \nabla_{\dot{x}} v \|_g^2 \right] = \mathbb{E}_{t,x_t} \left[ w \langle v_{\theta^-} - v_\theta + \dot{u}_{\theta^-}, \dot{u}_{\theta^-} \rangle_g \right], \tag{9}$$

where $\dot{u}_\theta := \dot{x} + \dot\kappa v_\theta + \kappa \nabla_{\dot{x}} v_\theta$. Here, we make one key approximation that $\mathrm{d}(\exp_x)_u \approx \mathrm{d}(\exp u)_x$ such that the second term can be combined with the first one. Also note that $\mathrm{d}(\exp_x)_u$ is a linear mapping that does not change the optimality of the loss. Therefore, we simply ignore such a transform and optimize the norm of $\dot{u}_\theta$ at $x_t$. Indeed, for the flat-torus and any manifold whose exponential map is symmetric, the identity $\mathrm{d}(\exp_x)_u = \mathrm{d}(\exp u)_x$ always holds. For general Riemannian manifolds, the following result holds:

**Proposition 3.1.** *For $u, v \in T_x\mathcal{M}$, if $u, v$ are parallel, then $\mathrm{d}(\exp_x)_u(v) = \mathrm{d}(\exp u)_x(v)$.*

*Proof.* As both differentials are linear, it suffices to verify $\mathrm{d}(\exp_x)_u(u) = \mathrm{d}(\exp u)_x(u)$. Consider the consistency parameterization $f(x_t, t) := \exp_{x_t}(1 - t)u_t$, where $u_t = \Pi_{x_0, x_t; \gamma}(u)$ is the parallel-transported tangent vector along the geodesic $\gamma$ from $x_0$ to $x_t$. Therefore, $f(x_t, t)$ is a constant-speed dynamics along the geodesics defined by point $x_0 = x$ and vector field $u_0 = u \in T_x\mathcal{M}$. As $f$ travels in constant speed, we have $f(x_t, t) \equiv \exp_x u, \forall t \in [0, 1]$. Taking the derivative with respect to $t$, we obtain:

$$\dot{f} = \mathrm{d}(\exp_x)_u \left(-u + (1 - t)\nabla_{\dot{x}} u\right) + \mathrm{d}(\exp u)_x (\dot{x}) = 0. \tag{10}$$

As $\dot{x} = u$, the covariant derivative $\nabla_{\dot{x}} u = \nabla_{\dot{x}} \dot{x}$ vanishes as $f$ traces a geodesic. Therefore, we have $\mathrm{d}(\exp_x)_u(u) = \mathrm{d}(\exp u)_x(\dot{x}) = \mathrm{d}(\exp u)_x(u)$, which concludes the proof. $\square$

In the original RCM loss, the two tangent vectors follow the predicted and the marginal vector field directions, respectively. This indicates that, if the pre-trained model approximates the marginal vector field well, the approximation shall be more accurate. The adaptation from RCD to RCT follows a similar procedure of marginalization in the proof for Theorem 3.2. We summarize the simplified RCD and RCT training procedure in Algorithm 1 and 2, where the key differences are highlighted in red. The original RCD and RCT follow similar training algorithms except for optimizing the loss in Eq.8. Sampling from RCM follows the same setup as the Euclidean CM, which is described in Appendix C.2.

---

**Algorithm 1** Simplified Riemannian Consistency Distillation (sRCD)

1: **Input:** Pre-trained RFM $s_\phi$.
2: **while** not converged **do**
3:     Sample data $x_1$, noise $x_0$, and $t$.
4:     Calculate $x_t = \exp_{x_1}(\kappa_t \log_{x_1}(x_0))$.
5:     Calculate $s_\phi(x_t, t)$.
6:     Calculate $\dot{u}_\theta = s + \dot\kappa v_\theta + \kappa \nabla_s v_\theta$.
7:     Optimize the loss with
        $\nabla_\theta w \langle v_{\theta-} - v_\theta + \dot{u}_{\theta-}, \dot{u}_{\theta-} \rangle_g$.
8: **end while**

**Algorithm 2** Simplified Riemannian Consistency Training (sRCT)

1: **Input:** None.
2: **while** not converged **do**
3:     Sample data $x_1$, noise $x_0$, and $t$.
4:     Calculate $x_t = \exp_{x_1}(\kappa_t \log_{x_1}(x_0))$.
5:     Calculate $\dot{x}_t = \dot\kappa_t \log_{x_t} x_1 / \kappa_t$.
6:     Calculate $\dot{u}_\theta = \dot{x} + \dot\kappa v_\theta + \kappa \nabla_{\dot{x}} v_\theta$.
7:     Optimize the loss with
        $\nabla_\theta w \langle v_{\theta-} - v_\theta + \dot{u}_{\theta-}, \dot{u}_{\theta-} \rangle_g$.
8: **end while**

---

### 3.3 Kinematics Perspective of Riemannian Consistency Model

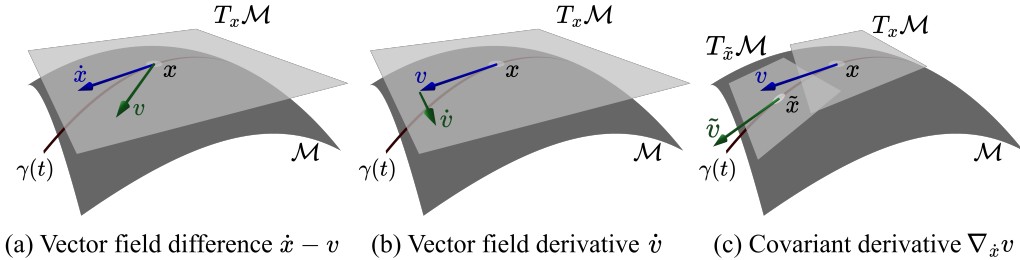

(a) Vector field difference $\dot{x} - v$    (b) Vector field derivative $\dot{v}$    (c) Covariant derivative $\nabla_{\dot{x}} v$

Figure 2: Three components of the variations in the consistency objective.

We now provide a perspective from kinematics that intuitively explains the terms in the RCM objective. Consider a point moving on the Riemannian manifold, the RCM objective essentially enforces the infinitesimal equilibrium on the target $f_\theta(x_t, t)$ along the PF-ODE of $x$. Locally, the change of the final target can be decomposed into three components:

- The *difference* in the predicted and marginal vector fields, Figure 2(a).
- The *intrinsic change* of the vector field, Figure 2(b).
- The *extrinsic change* of the vector field due to the geometric constraint, Figure 2(c).

The first and second terms are more intuitive, represented by the difference and derivative of the vector field. Also appearing in the Euclidean CM in [54]'s parameterization, these two terms can be directly generalized to any Riemannian manifold with the standard calculation in the tangent space $T_{x_t}\mathcal{M}$, a vector space where all vector calculations are valid with the additional Riemannian metric.

The third term, however, is unique for general Riemannian manifolds, as it involves the geometric properties of the manifold. Intuitively, the curved geometry leads to different tangent spaces at different points, where tangent vectors are not directly comparable. In order to differentiate the vector field at adjacent points, the *covariant derivative* is introduced as a generalization of the directional derivative in the Euclidean case. More concretely, the covariant derivative $\nabla_{\dot{x}}v$ describes how the vector field $v$ changes along the curve defined by $\dot{x}$. In other words, even if the vector field $v$ is "constant" along the curve, its time derivative is not necessarily zero; it is only when we consider such additional geometric properties that we can arrive at the results that the covariant derivative is indeed zero for a constant vector field. In this case, the vector field is called *parallel-transported* along the curve $\dot{x}$.

In this way, we have included all three components that affect the target position in the kinematics of the Riemannian manifold, with the last extrinsic change capturing the manifold's geometric properties. We now discuss some concrete examples of Riemannian manifolds to demonstrate how different RCMs can explicitly or implicitly learn the consistent kinematics on the manifold. Specifically, we focus on the covariant derivative $\nabla_{\dot{x}}$, which appears in all continuous-time RCM losses. The explicit mathematical formulae can be found in Appendix B.

**Euclidean Space.** The Euclidean space is a flat manifold with the canonical Euclidean metric $g_{ij} = \delta_{ij}$. The covariant derivative reduces to the common time derivative as $\nabla_{\dot{x}}v = \dot{v}$. This intuitively makes sense as the vector field can be trivially transported everywhere on the flat manifold. With the linear scheduler $\kappa_t = 1 - t$, we obtain the original vector field consistency model loss $\|\dot{x} - v + (1-t)\dot{v}\|$ in [54] as expected. A similar analysis holds for the flat torus $\mathbb{T}^n = (S^1)^n$, where $S^1$ is the 1D spherical manifold (see below).

**Spherical Manifold.** The $n$-sphere $S^n = \{\|x\| = 1 \mid x \in \mathbb{R}^{n+1}\}$ is a $n$-dimensional manifold which inherits the Euclidean metric of $\mathbb{R}^{n+1}$. The covariant derivative on sphere reads $\nabla_{\dot{x}}v = \dot{v} + \langle v, \dot{x}\rangle x = 0$. If $v = \dot{x}$ such that the first variation is perfectly optimized, the acceleration becomes $\dot{v} = -x\langle v, v\rangle$ with a magnitude of $\langle v, v\rangle = \|v\|^2$ and points in the inverse direction of $x$ towards the origin. This result coincides with the acceleration formula for uniform circular motion, which reads $\|a\| = \|v\|^2/r$, with the direction also pointing to the center. In this way, with the non-zero curvature, the covariant derivative provides additional geometry-aware information for the consistency objective.

**3D Rotation Group.** The 3D rotation group $SO(3)$ of all 3D rotation matrices is a Lie group with a Riemannian structure. With the group property, the tangent space at every point is isomorphic to the tangent space $\mathfrak{g} = T_eG$ at the identity element $e = I$, also known as the Lie algebra. The Lie algebra of $SO(3)$ is $\mathfrak{so}(3)$, the vector space of skew-symmetric matrices. Using the 3-vector representations, $SO(3)$ has a natural bi-invariant Riemannian metric $g(u, v) = \langle u, v\rangle$. The covariant derivative can be calculated as $\nabla_{\dot{x}}v = \dot{v} + \frac{1}{2}[\dot{x}, v]$, where $[u, v] = u \times v$ is the cross-product for 3-vectors. This formulation has a close relation to the rotating frame of reference in kinematics, where the additional cross product represents the Coriolis term induced by the non-inertial rotating reference frame. Furthermore, as $[v, v]$ vanishes, we have $\nabla_v v = \dot{v}$. Essentially, this condition indicates that $v$ should be a left-invariant vector field to trace a geodesic on $SO(3)$.

## 4 Experiments

To demonstrate the effectiveness of the RCM framework on Riemannian manifolds, we carry out extensive experiments on various non-Euclidean settings. In addition to Riemannian consistency distillation and training (**RCD**, **RCT**), we also test with the simplified objective in Eq.9 (**sRCD**, **sRCT**). Furthermore, to demonstrate the advantage of continuous-time RCM, we include the discrete-time RCD (**dRCD**) with a discretization step $\Delta t = 10^{-2}$. In principle, as all Riemannian manifolds

used in this work have a natural inclusion mapping to the ambient Euclidean space, the Euclidean CM can be directly applied. We call such a naive adaptation (on distillation) **CD$_\text{naive}$/CT$_\text{naive}$**, to distinguish from our RCD. We use a 2-step generation setup for all the models above during inference and set the fixed intermediate time step to $t = 0.8$ for all manifolds and datasets. As a reference, the results for the 100-step (**RFM-100**) and 2-step (**RFM-2**) Riemannian flow matching model are also provided.

Across all experiments on all manifolds, we employ the same network architecture, with the sole difference being the input and output dimensions that vary according to the manifold dimension. We use the linear scheduler $\kappa_t = 1 - t$ with the weighing function $w_t = (\dot{\kappa}_t/\kappa)^2 = t^2/(1-t)^2$ following [33]. Given the critical importance of stable Jacobian-vector products (JVP) $\dot{v} = \partial_t v + \dot{x}\partial_x v$ for training consistency models, we followed EDM2 [23] and utilized magnitude-preserving fully-connected (MP-FC) layers with force weight normalization. Additionally, we incorporated time information into the model using magnitude-preserving Fourier features and concatenation. The detailed network architecture and hyperparameters are provided in Appendix C.

To provide a model-agnostic evaluation metric for the generation quality, we use nonparametric clustering approaches of kernel density estimation (KDE) and maximum mean discrepancy (MMD) [18]. Specifically, for the 2-sphere datasets, we use KDE with the von Mises–Fisher kernel with a bandwidth of 0.02 and the haversine distance (geodesic distance on sphere) to estimate the spherical densities for the ground truth data and the generated samples of the same size. We then mesh-grid the sphere and calculate the Kullback–Leibler divergence (KLD) between the two kernel densities. For flat-tori and SO(3), the analogue of the standard Gaussian kernel is not well-defined, so we switch to MMD to measure distributional fitness. MMD is a kernel-based, distribution-free two-sample test, with a lower value indicating better distributional fitness. On the two sample sets, the MMD can be calculated as:

$$\text{MMD}^2(X, Y) = \frac{1}{n(n-1)}\sum_{i,j=1}^{n} k(x_i, x_j) + k(y_i, y_j) - 2k(x_i, y_j), \quad k(x, y) := \exp(-\gamma d_g^2(x, y)),$$

(11)

where the symmetric Gaussian-like kernel function $k$ is defined using the geodesic distance $d_g$ with the bandwidth parameter $\gamma = 1$.

### 4.1 Spherical Manifold

Table 1: KL divergence between the estimated kernel densities on the 2-sphere datasets. The dataset size is noted. Except for RFM-100, the best results are in **bold** and the second best are underlined.

| KLD↓ | Flow Matching | | Consistency Distillation | | | | Consistency Training | |
|---|---|---|---|---|---|---|---|---|
| | RFM-100 | RFM-2 | sRCD | RCD | dRCD | CD$_\text{naive}$ | sRCT | RCT |
| Earthquake $_{6,124}$ | 1.51 | 10.99 | **2.13** | 2.22 | 6.20 | 3.66 | 2.38 | 2.38 |
| Volcano $_{829}$ | 1.77 | 35.40 | **3.36** | 3.84 | 17.19 | 5.44 | 4.47 | 4.78 |
| Fire $_{4,877}$ | 0.53 | 9.79 | **1.65** | 1.71 | 8.01 | 3.39 | 1.74 | 1.72 |
| Flood $_{12,810}$ | 1.33 | 8.17 | 2.27 | 2.41 | 6.21 | 2.81 | 2.39 | **2.23** |

For our experiments on spherical manifolds, we utilize the real-world data comprising four distinct earth location datasets: volcanic eruptions [40], earthquakes [39], floods [4], and wildfires [46], collected by [36]. Following [7], we assume the Earth's surface to be a perfect sphere. For the calculation of KLD between estimated kernel densities, we sampled the same number of points as the dataset for each model.

As shown in Table 1 and Figure 3 on the Flood data, the 100-step RFM achieves the lowest KLD as expected. However, its performance degrades significantly with 2-step sampling. In contrast, all consistency methods markedly improved sample quality in the 2-step setting. Directly using a naive Euclidean consistency model loss yielded the poorest results, highlighting the necessity of RCM's covariant derivative formula to accurately characterize the manifold's geometry. Similar to the results in [50], the results from consistency distillation were superior to those from consistency training. Furthermore, the simplified loss we proposed streamlines the calculation of the differentials of the exponential map without sacrificing generation quality. As a result, the simplified loss yields even better results empirically.

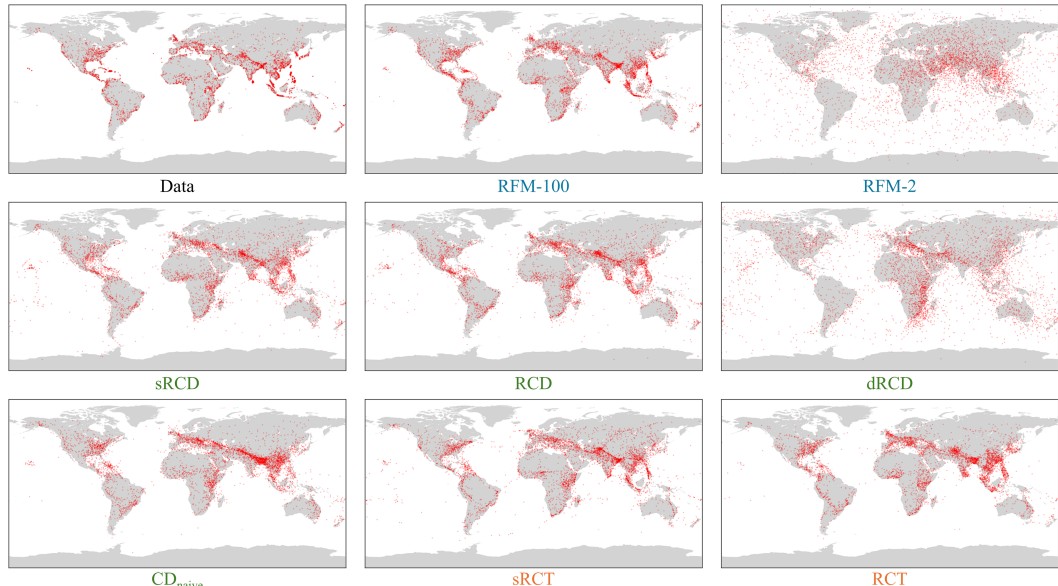

Figure 3: Generations on the Flood dataset on the 2-sphere. Except for RFM-100, all models use 2-step generation. The FM, CD, and CT models are colored in blue, green, and orange.

## 4.2 Flat Torus

We evaluate our RCM framework on flat tori using a synthetic checkerboard dataset[1] as well as pre-processed protein [32] and RNA [38] datasets, whose torsion angles can be represented on the 2D and 7D tori, respectively. Specifically, for the checkerboard data, we randomly generated 100k sampling points for training. The protein dataset contains 166,305 samples, and the RNA dataset contains 9,473 samples.

The MMD results are presented in Table 2, with additional visualizations in Appendix D.3. RFM with 100 sampling steps still achieves the best performance, while its performance is poor with only two sampling steps. Since the torus is flat, $d(\exp_x)_u, d(\exp u)_x$ are identity mappings such that the simplified loss is precisely

Table 2: Maximum mean discrepancy (MMD) scores on the 2D flat torus. Except for RFM-100, the best results are in **bold** and the second best are underlined.

| MMD↓/ $10^{-2}$ | Flow Matching | | Consistency Training | | | Consistency Distillation |
|---|---|---|---|---|---|---|
| | RFM-100 | RFM-2 | RCD | dRCD | CD$_{naive}$ | RCT |
| Board | 0.47 | 14.04 | 0.61 | **0.37** | 3.49 | 1.89 |
| Protein | 0.35 | 15.19 | 3.84 | **3.79** | 8.40 | 5.54 |
| RNA | 0.97 | 14.03 | 7.82 | **3.39** | 10.44 | 5.60 |

equivalent to the original loss. Surprisingly, the discrete version of RCD achieves decent results. We argue that this might be because the flat torus is more similar to a Euclidean space and less prone to numerical issues compared to other manifolds. Finally, the trivial Euclidean consistency model CD$_{naive}$ ignores the periodic nature of the torus, leading to boundary issues and significantly underperforming RCMs.

## 4.3 3D Rotation Group

For the generative modeling on the 3D Rotation Group $SO(3)$, we follow [30] to use synthetic rotation datasets with three different modes (Cone, Fisher, Line) on $SO(3)$. We additionally include a more challenging dataset that projects the 2D Swiss roll onto the $SO(3)$ manifold. For each dataset, 100k samples are generated for training the Riemannian flow matching and consistency models. During evaluation, 10k rotations are sampled for each model for MMD calculation.

---

[1]We refer to Flow Matching Guide and Code to implement our synthetic dataset.

Table 3: Maximum mean discrepancy (MMD) scores on the $SO(3)$ datasets. Except for RFM-100, the best results are in **bold** and the second best are underlined.

| MMD$\downarrow/10^{-2}$ | Flow Matching | | Consistency Distillation | | | | Consistency Training | |
|---|---|---|---|---|---|---|---|---|
| | RFM-100 | RFM-2 | sRCD | RCD | dRCD | CD$_{naive}$ | sRCT | RCT |
| Swiss Roll | 1.35 | 19.64 | 1.51 | **1.47** | 8.69 | 2.75 | 4.17 | 8.23 |
| Cone | 7.38 | 19.96 | 5.47 | 6.30 | 20.39 | 21.46 | 7.53 | **3.78** |
| Fisher | 4.02 | 17.41 | 5.81 | **5.71** | 16.41 | 6.87 | 8.59 | 7.00 |
| Line | 4.87 | 15.50 | 3.06 | **2.39** | 14.93 | 9.36 | 3.75 | 3.32 |

The quantitative results of MMD on $SO(3)$ are summarized in Table 3. Again, the simplified version of RCD and RCT achieves similar performance to the exact one. Noticeably, the discrete-time RCD performs worse on $SO(3)$, probably due to larger discretization errors on the curve geometry. The naive Euclidean CD also falls short in terms of MMD, even though the 3-vector representation allows for arbitrary rotation vectors as the output. Interestingly, the RCT model on the Cone dataset achieved the best MMD score, even better than the RCD counterpart. This is probably because the pre-trained RFM in this case is not as good as the other datasets, as demonstrated by a relatively high MMD for the RFM-100 model. In this way, the learned marginal vector field may not be accurate enough, leading to worse performance on the distillation approaches. The RCT model, on the other hand, does not suffer from this limitation, as it relies on the conditional vector fields instead.

## 4.4 Ablation Study

Scalability to higher-dimensional Riemannian manifolds and sampling efficiency are two crucial aspects for the generalization of RCM and other baseline models. To demonstrate scalability, we follow the setup in [27] on high-dimensional tori. The target distribution is a wrapped standard Gaussian distribution centered at the original, with the torus defined on $[0, 2\pi]^D$. Therefore, the high-density regions cover every corner of the hypercube, making it necessary to enforce the manifold constraint (periodic boundary condition). We estimate the Gaussian parameters based on maximum likelihood estimation, and calculate the Fréchet distance to the ground truth. 5k points are sampled as the training dataset, and the same number of samples is generated for parameter estimation. The results are shown in Table 4, where RCT consistently outperforms RFM and the naive Euclidean CM in the few-step setup for a manifold dimension up to 128. Specifically, we noted that the performance of CT$_{naive}$ drastically degrades on higher-dimensional manifolds. We hypothesize it is because the increasing manifold dimension makes it exponentially harder for the Euclidean model to comply with the manifold constraint, as the number of constraints grows exponentially. Our results demonstrate the scalability of RCM, further necessitating the need to respect manifold properties in consistency models.

Table 4: Fréchet distance (lower is better) between the ground truth Gaussian parameters and those estimated from the generations on high-dimensional tori. All samples are generated using 2 steps.

| Torus dim | 2 | 4 | 8 | 16 | 32 | 64 | 128 |
|---|---|---|---|---|---|---|---|
| RFM | 0.52 | 0.70 | 1.01 | 1.41 | 1.95 | 1.47 | 1.83 |
| RCT | **0.22** | **0.31** | **0.54** | **0.81** | **0.46** | **0.58** | **0.62** |
| CT$_{naive}$ | 0.73 | 2.69 | 1.58 | 2.16 | 2.41 | 24.80 | 35.16 |

We also provide a comprehensive study on the impact of the number of sampling steps on the generation quality for different models in Appendix D.1, where RCM variants consistently outperform the naive Euclidean baseline across all NFEs. The empirical benchmark on the sampling stage speedup is provided in Appendix D.2.

## 5 Related Work

**Riemannian Generative Models.** Modeling data distributions beyond Euclidean space is essential for applications like protein modeling [56] and geosciences [22]. [10, 20] successfully extend

diffusion models [19, 49] to Riemannian manifolds that learn to reverse the noising process. [31] introduced practical improvements for Riemannian diffusion models for certain Riemannian symmetric spaces to enhance scalability. [21] tackled scalability by employing bridge processes for generative diffusion on Riemannian manifolds. Another approach involves extending continuous normalizing flows [8, 17, 27] to manifolds. Early attempts leverage mapping between manifolds and Euclidean spaces [15] or focused on simple manifolds using maximum likelihood training [36]. [43] proposed simulation-free training for continuous flows, though they scaled mainly to high dimensions. [6] expanded flow matching [27, 28] and achieved simulation-free training for simple geometries. [51] introduced a one-step free-form flow generator for manifolds. Despite these advances, the development of a one-step Riemannian Generator remains largely unexplored, motivating our research in this area.

**One-Step Generator.**    Diffusion models, also known as score-based generative models [19, 47, 1, 49], along with flow models [27], have seen substantial success in various fields. However, these models typically necessitate hundreds of functional evaluations to ensure high-quality outputs. To address this, recent research has focused on reducing the number of function evaluations (NFEs) by adopting advanced solvers [34, 59], introducing better prior-data couplings [29, 52], and leveraging progressive distillation [37, 44]. There is growing interest in developing one-step generators capable of producing samples with a single function evaluation. Notably, some studies have successfully distilled multi-step diffusion models into single-step student generators [35, 57, 53, 45], often drawing from pre-trained teacher models. Another emerging approach involves consistency models [50, 54, 33], which learn a consistency function that maps noisy data along an ODE trajectory to the corresponding clean data. These models can be trained either by distillation or from scratch. The idea of "consistency" inspires subsequent works such as shortcut models [12].

# 6    Conclusion

In this work, we propose the Riemannian consistency model (RCM) as an extension to the existing consistency model on Euclidean domains. We have provided an explicit closed-form continuous-time RCM objective with manifold-specific operations. Utilizing the marginalization trick on the Riemannian manifold, we rigorously prove that Riemannian consistency training (RCT) is mathematically equivalent to Riemannian consistency distillation (RCD). We further propose a simpler objective without the need for the explicit form of the differentials of the exponential map while still achieving similar and even better empirical performance. We provide a novel perspective from kinematics that offers an intuitive interpretation of the RCM objective as geometry-aware motions.

Our RCM, together with its CD and CT variants, achieved significantly better generation quality than the vanilla flow matching and Euclidean consistency model in the few-step generation setup, demonstrating the necessity of incorporating intrinsic geometric information in the model design. We will continue to explore the possibility of scaling up Riemannian flow matching and hope to inspire more efficient flow model architectures in various downstream domains like protein design.

## Acknowledgement

This work was funded by the DOE Center for Advanced Bioenergy and Bioproducts Innovation (U.S. Department of Energy, Office of Science, Biological and Environmental Research Program under Award Number DE-SC0018420).

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

# Supplementary Material

## A   Riemannian Geometry

In this section, we give a more detailed mathematical background on Riemannian geometry. A more comprehensive background on the Riemannian manifold can be found in standard mathematical textbooks like [13].

A Riemannian manifold $\mathcal{M}$ is a real, smooth manifold equipped with a positive definite inner product $g$ on the tangent space $T_x\mathcal{M}$ at each point $x \in \mathcal{M}$. Let $T\mathcal{M} = \bigcup_{x \in \mathcal{M}} \{x\} \times T_x\mathcal{M}$ be the *tangent bundle* of the manifold $\mathcal{M}$, a time-dependent *vector field* on $\mathcal{M}$ is a mapping $u_t : [0,1] \times \mathcal{M} \to T\mathcal{M}$ where $u_t(x) \in T_x\mathcal{M}$. A *geodesic* is a locally distance-minimizing curve on the manifold. The existence and the uniqueness of the geodesic state that for any point $x \in \mathcal{M}$ and for any tangent vector $u \in T_x\mathcal{M}$, there exists a unique geodesic $\gamma : [0,1] \to \mathcal{M}$ such that $\gamma(0) = x$ and $\dot{\gamma}(0) = u$. The *exponential map* $\exp : \mathcal{M} \times T\mathcal{M} \to \mathcal{M}$ is uniquely defined to be $\exp_x(u) := \gamma(1)$. The *logarithm map* $\log : \mathcal{M} \times \mathcal{M} \to T\mathcal{M}$ is defined as the inverse mapping of the exponential map such that $\exp_x(\log_x y) \equiv y, \forall x, y \in \mathcal{M}$.

Let $\Gamma(T\mathcal{M})$ denote the space of vector fields on $\mathcal{M}$, a *covariant derivative* or *affine connection*

$$\nabla : \Gamma(T\mathcal{M}) \otimes \Gamma(T\mathcal{M}) \to \Gamma(T\mathcal{M}), \tag{12}$$
$$(u, v) \mapsto \nabla_u v$$

is a bilinear map on $\mathcal{M}$ that satisfies:

1. $\nabla$ is tensorial in the first argument:

$$\nabla_{u_1+u_2} v = \nabla_{u_1} v + \nabla_{u_2} v, \quad \forall u_1, u_2, w \in \Gamma(T\mathcal{M}), \tag{13}$$
$$\nabla_{fu} v = f\nabla_u v, \quad \forall f \in C^\infty(\mathcal{M}), u, v \in \Gamma(T\mathcal{M}). \tag{14}$$

2. $\nabla$ is $\mathbb{R}$-linear in the second argument:

$$\nabla_u(v_1 + v_2) = \nabla_u v_1 + \nabla_u v_2, \quad \forall u, v_1, v_2 \in \Gamma(T\mathcal{M}), \tag{15}$$

and it satisfies the Leibniz rule:

$$\nabla_u(fv) = v(f)u + f\nabla_u v, \quad \forall f \in C^\infty(\mathcal{M}), u, v \in \Gamma(T\mathcal{M}). \tag{16}$$

The concept of the affine connection generalizes the idea of directional derivatives in the Euclidean case and allows for the differentiation of vector fields at different points. Consider a smooth curve $\gamma_t$, a vector field $u$ is said to be *parallel-transported* along the curve $\gamma$ if $\nabla_{\dot{\gamma}} u = 0$. Intuitively, such a condition indicates the vector field $u$ remains "constant" along the curve $\gamma$. We call the mapping $\Pi_{x_0, x_t; \gamma} : T_{x_0}\mathcal{M} \to T_{x_t}\mathcal{M}, u_0 \mapsto u_t$ the *parallel transport* of the tangent vector $u_0$ along the curve $\gamma$. Furthermore, a curve $\gamma$ is said to be *autoparallel* if $\nabla_{\dot{\gamma}} \dot{\gamma} = 0$. Intuitively, a point along such a curve travels at "constant" speed as evaluated by the affine connection.

A *Levi-Civita connection* is the unique affine connection that is torsion-free and metric compatible $\nabla g = 0$. The covariant derivative can be explicitly expanded as

$$\nabla_u v = \dot{v}^k + \Gamma_{ij}^k v^i u^j, \tag{17}$$

where $\Gamma_{ij}^k$ are the Christoffel symbols that can be calculated explicitly using the Riemannian metric $g_{ij}$ and the inverse metric $g^{ij}$ as:

$$\Gamma_{ij}^k = \frac{1}{2} g^{km}(\partial_i g_{mj} + \partial_j g_{mi} - \partial_m g_{ij}). \tag{18}$$

In this way, the Christoffel symbols serve as the geometry-aware terms that correct the time derivative of the vector field in the Euclidean case. One important property of the Levi-Civita connection is that every geodesic $\gamma$ is autoparallel:

$$\nabla_{\dot{\gamma}} \dot{\gamma} = 0. \tag{19}$$

In this way, the geodesic equation in Eq.19 can be written locally as:

$$\ddot{\gamma}^k + \Gamma_{ij}^k \dot{\gamma}^i \dot{\gamma}^j = 0, \quad k = 1, \ldots, n. \tag{20}$$

The *differentials of the exponential map* $\mathrm{d}(\exp_x)_u, \mathrm{d}(\exp u)_x : T_x\mathcal{M} \to T_y\mathcal{M}, y = \exp_x u$ are linear mappings between tangent spaces. For $\mathrm{d}(\exp_x)_u$, we fix $x$ and evaluate the differential with respect to the tangent vector at $u$; for $\mathrm{d}(\exp u)_x$, we fix $u$ and evaluate the differential with respect to the base point at $x$. Intuitively, these differentials describe how a small change at $x$ will affect the change at $y = \exp_x u$. Note that, from the parallel transport equation in Eq.17, we can obtain the approximation of infinitesimal parallel transport along the direction of $\mathrm{d}x$ as

$$\mathrm{d}u^k = -\Gamma_{ij}^k u^i (\mathrm{d}x)^j. \tag{21}$$

Such a result indicates that, for non-flat Riemannian manifolds with non-zero Christoffel symbols, there exists an acceleration for enforcing the manifold constraint even if the vector field is "constant" (more precisely, parallel transported along the curve). This is especially important for the calculation of $\mathrm{d}(\exp u)_x$, where an additional term involving $\mathrm{d}(\exp_x)_u$ needs to be added to account for the extrinsic change in the vector field because of the base point change.

## B  Geometry on Specific Riemannian Manifold

In this section, we further provide additional information on the geometric properties of the Riemannian manifolds used in this work.

### B.1  Euclidean Space and Flat Torus

The Euclidean space with the canonical Euclidean inner product forms a Riemannian structure. As expected, the geodesic distance is simply the Euclidean distance. As the manifold is flat, the exponential and logarithm maps simply read:

$$\exp_x u = x + u, \tag{22}$$
$$\log_x y = y - x. \tag{23}$$

All Christoffel symbols vanish on the Euclidean space, so the covariant derivative $\nabla_{\dot{x}} v = \dot{v}$ reduces to the ordinary time derivative. Direct calculation gives $\mathrm{d}(\exp_x)_u(v) = \mathrm{d}(\exp u)_x(v) = v$ are the identity map. This makes sense as the tangent vectors can be parallel transported freely anywhere in the flat space.

The flat torus $\mathbb{T}^n = (S^1)^n$ can be viewed as the quotient manifold of the Euclidean space by identifying $x + 2k\pi \sim x$ for $k \in \mathbb{Z}$ along each dimension. Therefore, it is also a flat manifold that inherits most properties of the Euclidean space. The only difference is that, when calculating the exponential and logarithm maps, we shall follow the minimum-image convention along each dimension and wrap the results back to the interval of $[0, 2\pi]$.

### B.2  Spherical Manifold

The $n$-sphere $S^n$ is an $n$-dimensional Riemannian manifold that inherits the canonical inner product from $\mathbb{R}^{n+1}$ as

$$\langle u, v \rangle_S = \langle u, v \rangle = \sum_{i=1}^n u_i v_i, \quad u, v \in T_x S^n. \tag{24}$$

The tangent space $T_x S^n = \{u | \langle u, x \rangle = 0\}$ is a $n$-dimensional hyperplane perpendicular to the vector $x$. The geodesic on the sphere follows the great circle between two points, and the geodesic distance can be calculated as

$$d_S(x, y) = \arccos\langle x, y \rangle. \tag{25}$$

The exponential and logarithm maps can be calculated as:

$$\exp_x u = x \cos \|u\| + \frac{u}{\|u\|} \sin \|u\|, \tag{26}$$

$$\log_x y = \frac{\arccos(\langle x, y \rangle)}{\sqrt{1 - \langle x, y \rangle^2}} (y - x - \langle x, y - x \rangle x). \tag{27}$$

The embedded Christoffel symbols read $\Gamma_{ij}^k = x^k \delta_{ij}$, and the covariant derivative can be calculated as $\nabla_{\dot{x}} v = \dot{v} + \langle v, \dot{x} \rangle x$. This indicates that a small perturbation $\mathrm{d}x$ on the point will lead to a change of $-\langle v, \mathrm{d}x \rangle x$ on the tangent vector $v$.

Direct differentiation on Eq.26 gives:

$$d(\exp_x)_u(v) = v_{\parallel} \cos \|u\| + \frac{\sin \|u\|}{\|u\|}(v_{\perp} - \langle u, v \rangle x), \tag{28}$$

where $v_{\parallel} = \langle u, v \rangle u / \|u\|^2$ is the parallel component of $v$ with respect to $u$ and $v_{\perp} = v - v_{\parallel}$ is the orthogonal component. Similarly, for $d(\exp u)_x$ we have:

$$d(\exp u)_x(v) = v \cos \|u\| - \frac{\sin \|u\|}{\|u\|} \langle u, v \rangle x. \tag{29}$$

The first term is the direct differentiation result. The second term is exactly $d(\exp_x)_u(-\langle u, v \rangle x)$, which arises because, as mentioned above, a change in $x$ will lead to an additional change in $u$ in the covariant derivative. One can verify that, for any $v \in T_x S^n$, we have $\langle d(\exp_x)_u(v), \exp_x u \rangle = \langle d(\exp u)_x(v), \exp_x u \rangle = 0$.

### B.3   3D Rotation Group

The 3D rotation group $\mathrm{SO}(3)$ is a 3-dimensional Riemannian manifold with the Lie group structure. Usually, it can be considered as the group of all 3D rotation matrices $\mathrm{SO}(3) = \{R \in \mathbb{R}^{3\times3} : R^\top R = I, \det(R) = 1\}$ [55, 3], with it Lie algebra $\mathfrak{so}(3)$ consisting of all 3D skew-symmetric matrics $\mathfrak{so}(3) = \{A \in \mathbb{R}^{3\times3} : A = -A^\top\}$. However, it is worth noting that other formalisms can also be adopted, e.g., the quaternion representation [58]. In this work, we use the 3-vector representation (i.e., rotation vectors or axis-angle representations) for easy derivative calculation. Mathematically, define the vee-hat isomorphism as

$$(\hat{\cdot}) : \mathbb{R}^3 \to \mathfrak{so}(3), \begin{pmatrix} a_1 \\ a_2 \\ a_3 \end{pmatrix} \mapsto \begin{pmatrix} 0 & -a_3 & a_2 \\ a_3 & 0 & -a_1 \\ -a_2 & a_1 & 0 \end{pmatrix}, \tag{30}$$

$$(\check{\cdot}) : \mathfrak{so}(3) \to \mathbb{R}^3, \begin{pmatrix} 0 & -a_3 & a_2 \\ a_3 & 0 & -a_1 \\ -a_2 & a_1 & 0 \end{pmatrix} \mapsto \begin{pmatrix} a_1 \\ a_2 \\ a_3 \end{pmatrix}. \tag{31}$$

Then, the 3-vector representations are related to rotation matrices via the following Lie exponential map $\mathrm{Exp} : \mathfrak{so}(3) \to \mathrm{SO}(3)$ and Lie logarithm map $\mathrm{Log} : \mathrm{SO}(3) \to \mathfrak{so}(3)$:

$$R = \mathrm{Exp}(\hat{\theta}) = I + \frac{\sin \|\theta\|}{\|\theta\|}\hat{\theta} + \frac{1 - \cos \|\theta\|}{\|\theta\|^2}\hat{\theta}^2, \tag{32}$$

$$\hat{\theta} = \mathrm{Log}(R) = \frac{\|\theta\|}{2 \sin \|\theta\|}(R - R^\top), \|\theta\| = \arccos \frac{\mathrm{Tr}(R) - 1}{2}. \tag{33}$$

For simplicity, we will omit the vee-hat isomorphism notation and allow the Lie exponential map to apply to a 3-vector $R = \mathrm{Exp}\,\theta$ as if the hat has been applied, and also allow the Lie logarithm map to return a 3-vector $\theta = \mathrm{Log}\,R$ as if the vee has been applied. The Riemannian exponential and logarithm maps $\exp, \log : \mathbb{R}^3 \times \mathbb{R}^3 \to \mathbb{R}^3$ can be expressed using the Lie exponential and logarithm maps as

$$\exp_x u = \mathrm{Log}(\mathrm{Exp}\,x\,\mathrm{Exp}\,u), \tag{34}$$
$$\log_x y = \exp_{(-x)} y = \mathrm{Log}(\mathrm{Exp}(-x)\,\mathrm{Exp}\,y). \tag{35}$$

Here, we follow the body-frame convention for the rotation vectors such that the tangent vector $u$ can be interpreted as the local angular velocity relative to the global frame. Using the 3-vector, the canonical bi-invariant Riemannian metric is given by $g(u, v) = \langle u, v \rangle$, which is equivalent to $g(A, B) = \frac{1}{2}\mathrm{Tr}(A^\top B), A, B \in \mathfrak{so}(3)$. The geodesic distance then reads

$$d(x, y) = \|\log_x y\| = \|\mathrm{Log}(\mathrm{Exp}(-x)\,\mathrm{Exp}\,y)\|. \tag{36}$$

The Christoffel symbols $\Gamma_{ij}^k = \varepsilon_{kij}$ are the Levi-Civita symbols, and the covariant derivative can be calculated as $\nabla_{\dot{x}} v = \dot{v} + \frac{1}{2}[\dot{x}, v]$, where for 3-vector representations, the Lie bracket becomes the cross product $[u, v] = u \times v$.

For the calculation of the differentials, the left- and right-Jacobians $J_L(\theta), J_R(\theta)$ relate small perturbations in the 3-vectors. They are defined as:

$$J_L(\theta) := \sum_{n=0}^{\infty} \frac{1}{(n+1)!} \left(\hat{\theta}\right)^n, \quad J_R(\theta) := \sum_{n=0}^{\infty} \frac{1}{(n+1)!} \left(-\hat{\theta}\right)^n. \tag{37}$$

Specifically, we have $J_R(\theta) = J_L(-\theta) = J_L(\theta)^\top = R^\top J_L(\theta)$, where $R = \text{Exp}(\theta)$. The Baker-Campbell-Hausdorff (BCH) formula on SO(3) reads:

$$\text{Exp}(\theta + d\theta) \approx \text{Exp}\,\theta\,\text{Exp}(J_R(\theta)\,d\theta), \tag{38}$$

$$\text{Exp}(\theta + d\theta) \approx \text{Exp}(J_L(\theta)\,d\theta)\,\text{Exp}\,\theta, \tag{39}$$

$$\text{Log}(\text{Exp}\,\theta\,\text{Exp}\,d\theta) \approx \theta + J_R^{-1}(\theta)\,d\theta, \tag{40}$$

$$\text{Log}(\text{Exp}\,d\theta\,\text{Exp}\,\theta) \approx \theta + J_L^{-1}(\theta)\,d\theta. \tag{41}$$

With the BCH formula, we can obtain the following approximations of the Riemannian exponential and logarithm maps:

$$\log_y(y + dy) \approx J_R(y)dy, \tag{42}$$

$$\exp_x(u + du) \approx J_R^{-1}(y)J_R(u)du, \, y = \exp_x u, \tag{43}$$

$$\exp_{x+dx}(u) \approx J_L^{-1}(y)J_L(u)dx, \, y = \exp_x u. \tag{44}$$

Combining the above approximations, we have

$$d(\exp_x)_u(v) = J_R(u)v. \tag{45}$$

Similarly, we can calculate $d(\exp u)_x$ using the above approximations with the additional covariant derivative term as:

$$d(\exp u)_x(v) = R_u^{-1} J_R(x)v - \frac{1}{2} J_R(u)[v, u], \tag{46}$$

where $R_u$ denotes the rotation by the rotation vector $u$, and the second term again arises from the covariant derivative. The time derivative of the model using the Jacobian-vector product (JVP) should also be modified as:

$$\frac{dv}{dt} = \frac{\partial v}{\partial x} J_R^{-1}(x)\dot{x} + \frac{\partial v}{\partial t}, \tag{47}$$

where the additional left inverse Jacobian comes from the expansion of the reference vector field $\dot{x}$ using the right expansion BCH formula in Eq.40.

For any $v \in \mathbb{R}^3$, the Jacobians and their inverse can be calculated in closed form as:

$$J_L(u)v = J_R(-u)v = v + \frac{1 - \cos\|u\|}{\|u\|^2}[u, v] + \frac{\|u\| - \sin\|u\|}{\|u\|^3}[u, [u, v]], \tag{48}$$

$$J_L^{-1}(u)v = J_R^{-1}(-u)v = v - \frac{1}{2}[u, v] + \left(\frac{1}{\|u\|^2} - \frac{1 + \cos\|u\|}{2\|u\|\sin\|u\|}\right)[u, [u, v]]. \tag{49}$$

For any $v \in \mathbb{R}^3$, the application of a rotation vector can be calculated using the vector-form Rodrigues' formula:

$$R_u v = v + \frac{\sin\|u\|}{\|u\|}[u, v] + \frac{1 - \cos\|u\|}{\|u\|^2}[u, [u, v]]. \tag{50}$$

We elaborate why the formulae for SO(3) will incur additional terms and transformations. Specifically, one may notice a seemingly contradiction between the Taylor expansion we used in our proof of Theorem 3.1 of $\log_x y \approx y - x$ and the expansion for SO(3) in Eq.42. This is because we use a non-canonical representation for the rotations (*rotation vectors* instead of *rotation matrices*), effectively adding an additional coordinate transformation that leads to the Jacobians. By definition, a smooth Riemannian manifold resembles Euclidean space everywhere locally; therefore, the first-order Taylor expansion of the exponential map $\exp_x u \approx x + u$ always holds. However, by considering a smooth coordinate mapping $\varphi : \mathcal{M} \to \mathbb{R}^n$, the Jacobian of such a map $J_\varphi : T\mathcal{M} \to \mathbb{R}^n$ needs to be composed during the derivative calculation. Furthermore, for a non-Abelian Lie group, one must define both the left- and right-Jacobians. Indeed, the above procedure of calculating the Jacobians can be generalized to arbitrary smooth mappings $\varphi$, and the other canonical coordinates we used for the sphere and torus can be thought of as adopting the identity mapping with the identity Jacobian $J_\varphi = \text{id}$. Under the assumption of a canonical coordinate, our theoretical result in Theorem 3.1 safely holds for all smooth Riemannian manifolds.

## C  Experimental Detail

In this section, we provide comprehensive details on the model parameterization, training setup, and additional techniques we used to improve the performance of RCM.

### C.1  Model Parameterization

We found that the magnitude-preserving design principle of EDM2 [23] is very effective in stabilizing the JVP computation. However, since our task and datasets are relatively small (compared to the image generation task), we rebuilt a simple magnitude-preserving multilayer perceptron (MLP) using magnitude-preserving EDM2 modules.

**Magnitude-preserving fully-connected layer**  A fully-connected layer with input activation $\mathbf{x}$ and output activation is calculated as follows:

$$\text{MP-FC}(x) = \frac{w}{\|w\|_2} x \tag{51}$$

**Magnitude-preserving Fourier features**  The Fourier features are scaled by $\sqrt{2}$ using a cosine function as follows:

$$\text{MP-Fourier}(t) = \begin{bmatrix} \sqrt{2}\cos(2\pi(f_1 t + \varphi_1)) \\ \sqrt{2}\cos(2\pi(f_2 t + \varphi_2)) \\ \vdots \\ \sqrt{2}\cos(2\pi(f_N t + \varphi_N)) \end{bmatrix} \tag{52}$$

**Magnitude-preserving SiLU**  The SiLU nonlinear function should also be scaled to maintain the magnitude as follows:

$$\text{MP-SiLU}(x) = \frac{x}{0.596 \cdot (1 + e^{-x})} \tag{53}$$

**Magnitude-preserving Concatenation and Sum**  The concatenation of two input activations $x$ and $y$, scaled by constants $\omega_x$ and $\omega_y$, with a blend factor $a \in [0, 1]$ to adjust the balance between $x$ and $y$ is calculated as follows:

$$\text{MP-Cat}(x, y, a) = \sqrt{\frac{N_x + N_y}{(1-a)^2 + a^2}} \cdot \left[ \frac{1-a}{\sqrt{N_x}} x \oplus \frac{a}{\sqrt{N_y}} y \right] \tag{54}$$

where, $N_x$ and $N_y$ is the size of $\mathbf{x}$ and $\mathbf{y}$ in concatenation dimension. A similar formula is used for the sum operation as follows:

$$\text{MP-Sum}(x, y, a) = \frac{(1-a)x + ay}{\sqrt{(1-a)^2 + a^2}} \tag{55}$$

**Magnitude-preserving Block**  By using the above operations, we design a simple multilayer perceptron and residual connections to fuse noisy data and time embedding. Concretely, for inputs $x$ and $\text{emb}$, we perform the following computation:

$$h_x = \text{MP-FC}(\text{MP-SiLU}(\text{Norm}(x))) \tag{56}$$
$$h_{\text{emb}} = \text{MP-FC}(\text{emb}) + 1 \tag{57}$$
$$y = \text{MP-Sum}(\text{Norm}(x), \text{MP-FC}(h_x) \cdot h_{\text{emb}}, 0.3) \tag{58}$$

**Model Architectural**  We use MP-Blocks to build our model. First, we stack the blocks in the encoder and gradually increase the hidden dimension. Subsequently, in the decoder, we use the same blocks and concatenate the output of the corresponding layer of the encoder, similar to what UNet does. The time $t$ is encoded using MP-Fourier and passed through an MP-FC, which is subsequently fed into each block.

**Hyperparameter** In all our experiments, we use 4 blocks, each with dimensions [256, 512, 512, 256], and the dimension of time embedding is 256. For flow matching, we use a learning rate of $10^{-3}$, while for the consistency model, we use a learning rate of $10^{-4}$ with the Adam optimizer. The batch size varies depending on the dataset size and is typically 512 or 1024. We do not use dropout, nor do we employ other tricks such as learning rate decay. We train our model using a total of 50 million data samples.

All experiments were carried out on a single A100. The maximum GPU memory consumed is only around 5GB, which can be easily fit into GPUs with smaller memory. As we fixed the number of iterations, the entire training of each variant of CM models took approximately 4 hours, regardless of the dataset size.

## C.2 Model Sampling

We described the Riemannian flow matching sampling and our Riemannian consistency model sampling in Algorithm 3 and 4, respectively. The sampling algorithm for RCM is consistent with [50] on Euclidean cases. In each sampling step, the model first makes a prediction of the denoised results with the learned vector field (consistency parameterization in Eq.3). A geodesic interpolation step then follows to bring the prediction back to intermediate noisy data if there is more than one sampling step.

---

**Algorithm 3** RFM Sampling

---

1: **Input:** Trained RFM $v_\theta$, number of steps $N$.
2: Sample noise $x_0$.
3: **for** $t$ **in** $0, 1/N, 2/N, \ldots, 1 - 1/N$ **do**
4: $\quad x_{t+1/N} = \exp_{x_t}(-\dot{\kappa}_t v_\theta(x_t, t)/N)$. $\qquad\qquad\qquad\qquad$ ▷ Euler step
5: **end for**
6: **Return:** $x_1$.

---

**Algorithm 4** RCM Sampling

---

1: **Input:** Trained RFM $v_\theta$, number of steps $N$.
2: Sample initial noise $x_0$.
3: **for** $t$ **in** $0, 1/N, 2/N, \ldots, 1 - 1/N$ **do**
4: $\quad x_1 = \exp_{x_t}(\kappa_t v_\theta(x_t, t))$. $\qquad\qquad\qquad$ ▷ Consistency parameterization
5: $\quad$ Sample noise $x_0$.
6: $\quad x_{t+1/N} = \exp_{x_1}(\kappa_{t+1/N} \log_{x_1} x_0)$. $\qquad\qquad$ ▷ Interpolation to the next time step
7: **end for**
8: **Return:** $x_1$.

---

## C.3 Training Technique

**Tangent Warmup** In line with the findings of Lu and Song [33], we identified the covariant derivative term in our loss as a likely contributor to JVP instability. Consequently, we implemented a strategy of incrementally adding this term throughout the training process, detailed below:

$$\dot{f} = d(\exp_x)_u (\dot{\kappa}v) + d(\exp u)_x (\dot{x}) + r \cdot d(\exp_x)_u (\kappa \nabla_{\dot{x}} v) \tag{59}$$

where $r$ linearly increases from 0 to 1 over the first 10k training iterations. We made an interesting observation, not previously noted in Lu and Song [33], regarding the parameter $r$. Specifically, when $r = 0$, the loss function degenerates into a flow matching loss. As $r$ progressively increases, the model gradually transitions from a flow matching objective towards that of a consistency model. This progression is analogous to the methodology in ECT [16], which involves a reduction of $dt$ in discrete time steps. This insight offers a unified explanation for these two approaches, despite their different conceptual starting points.

**Tangent Clipping** To further stabilize the gradient variance in consistency models, we build upon the concept of tangent normalization proposed in Lu and Song [33]. We observed that when the

tangent norm is small, standard normalization preserves only directional information, which can impede further optimization. Therefore, drawing inspiration from gradient clipping, we introduce a 'tangent clipping' mechanism, detailed below.

$$\text{clip}(\dot{f}) = \dot{f} \cdot \max(M/\|\dot{f}\|, 1) \tag{60}$$

where $M$ represents the maximum tangent norm, which we set to 1 in our experiments. This approach is designed to maintain low gradient variance without inappropriately rescaling the tangent when its norm is small.

### C.4 Evaluation Details

For the estimation of kernel density on the 2-sphere, we use the off-the-shelf implementation from Scikit-Learn[2] with the haversine distance and the von Mises-Fisher kernel to match the spherical manifold. We follow [36] to choose the bandwidth of 0.02 and use a meshgrid of 90 longitudes and 90 latitudes for calculating the empirical KLD. For MMD calculation, we always use a bandwidth of 1 and use the geodesic distance in the exponential kernel to match the corresponding manifold.

For the scalability experiment on high-dimensional tori $[0, 2\pi]^D$, we first estimate the (diagonal) wrapped Gaussian using the maximum likelihood estimation (MLE) formula as

$$\hat{\mu} = \arctan\left(\sum_i \sin x_i \Big/ \sum_i \cos x_i\right), \tag{61}$$

$$\hat{\sigma}^2 = -\log\left(\frac{N}{N-1}\left(\frac{1}{N}\sum_i \sin x_i\right)^2 + \frac{N}{N-1}\left(\frac{1}{N}\sum_i \cos x_i\right)^2 - \frac{1}{N-1}\right), \tag{62}$$

where the index $i$ is over the data points. The above formula is applied to each manifold dimension (assuming a diagonal wrapped Gaussian). We then calculate the Fréchet distance (FD) to the ground truth standard Gaussian as:

$$\text{FD}^2 = \sum_j \hat{\mu}_j^2 + (\hat{\sigma}_j - 1)^2, \tag{63}$$

where the index $j$ is over the manifold dimension.

## D  Additional Results

In this section, we provide additional experimental results to further support the effectiveness, efficiency, and scalability of RCM.

### D.1  Impact of NFEs

We further provide a comprehensive study on the impact of the number of function evaluations (NFEs) on the final generation quality. We use the same model checkpoints trained on the earthquake dataset and alter only the sampling steps in $\{2, 5, 10, 20, 50, 100\}$ as the NFEs. The results are summarized in Figure 4. We have the following interesting observations:

- For the RFM model, increasing the NFEs has a significant impact on the generation quality, as the learned marginal vector fields are not necessarily straight, leading to significant errors in the few-step generation setup.

- RCD and RCT generations are more stable with respect to NFEs, consistently achieving decent generation quality. This indicates RCM can indeed shortcut the probability path on the Riemannian manifold.

- RCM variants consistently outperform the baselines, especially the naive Euclidean CM approach, demonstrating the necessity of Riemannian constraints. For RFM, the two lines intersect at around 20 NFEs, below which the RFM generation quality drastically drops.

---

[2]https://scikit-learn.org/

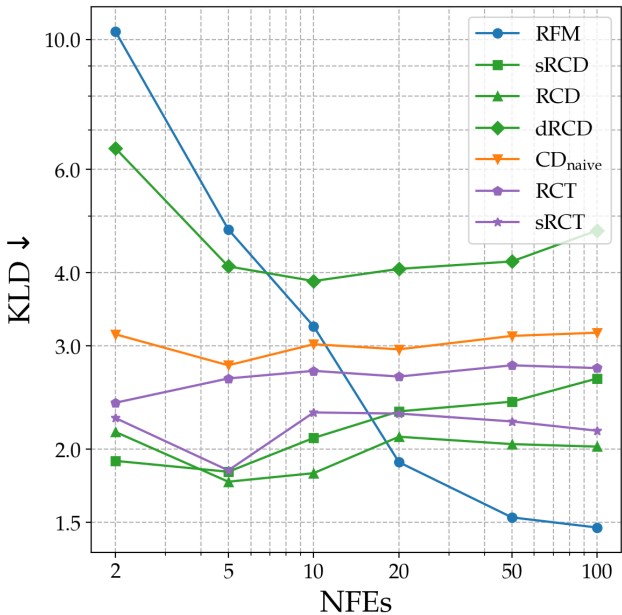

Figure 4: KLD vs NFEs for different models on the Earthquake dataset. RCD variants are colored in green, and RCT variants are colored in purple.

It shall be noted that the speed-quality tradeoff for the standard flow matching model (as demonstrated in the blue RFM curve) does not generally apply to CMs. Our NFE vs performance results generally echo the findings in Kim et al. [24], in which high NFEs may instead lead to poorer generations. In practice, the improvement in terms of KLD is also less significant for all CMs. It is also possible to adapt the techniques in Kim et al. [24] for Riemannian manifolds, resulting in finer-grained control over the sampling stage, which we leave for future work.

### D.2    Practical Time Complexity and Speedup

Intuitively, as the training and sampling procedure of RCM follows the Euclidean consistency model, the training time should be approximately the same while enjoying a speedup of the ratio of the sampling steps (NFEs) needed. In this way, the 2-step setup will incur a 50x speedup during inference. To provide more concrete quantitative results, we benchmarked the training stage time of RCM and sRCM compared to the Euclidean version ($CD_{naive}$) and the sampling stage speedup of two-step RCM compared to the 100-step RFM. The results are summarized in Table 5.

Table 5: Training time overhead compared to $CD_{naive}$ and sampling time speedup for RCM-2 compared to RFM-100 on three different datasets.

| Dataset | Earth | RNA | SO(3) |
|---|---|---|---|
| RCM train overhead | +5.9% | +<0.1% | +10.7% |
| sRCM train overhead | +4.1% | +<0.1% | +7.8% |
| RCM-2 sample speedup | ×46.5 | ×48 | ×43.2 |

For training, it is clear that even with a fairly small model, the additional Riemannian operations have minimal overhead in general. It is also expected that the overhead for SO(3) is larger than the sphere, as the former requires more operations, like the calculation of left or right Jacobians. Similarly, the simplified loss leads to fewer overheads as it does not require the calculation of the differentials of the exponential map. Generally, the overheads for RCm are still within a 10% range and will be even smaller for larger models, as the Riemannian operators only scale linearly with the data dimension but are model-independent. For sampling, the speedups are slightly below 50, probably due to the additional Riemannian operators and noising sampling in Algorithm 4. Despite this, the speedup

is drastic, demonstrating the effectiveness of RCM variants in few-step generation scenarios and offering a perfect solution for sampling efficiency.

## D.3  Additional Visualization

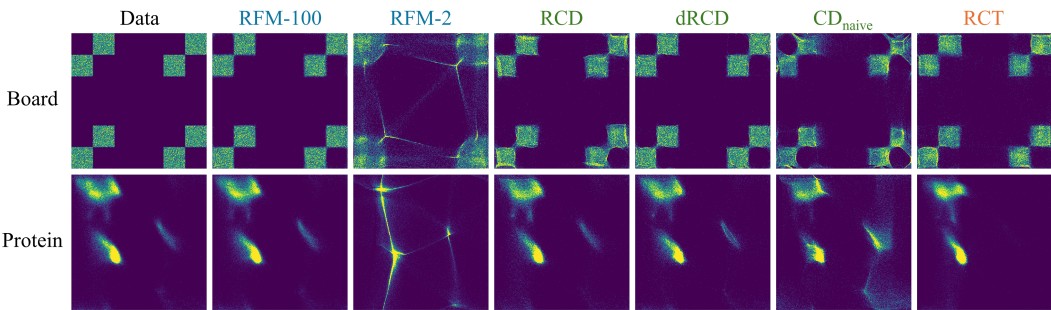

Figure 5: Generations on the 2D flat torus. The FM, CD, and CT models are colored in blue, green, and orange.

For the 2D torus datasets, we provide visualizations of the generations as the heat maps in Figure 5. Similar trends can be observed comparing different baselines. Interestingly, although the discrete-time RCD model achieved a better MMD score on the protein dataset, its visual results are not as good as RCD, as there are noticeable artifacts on the minor mode on the right in the dRCD generation.

For the spherical manifold, we provide generated points on all four datasets across all models in Figure 6. The number of generated samples is exactly the same as the corresponding ground truth data for fair comparison. It can be clearly seen that, while RFM-2 could still capture some coarse-grain features of the overall distribution, it failed to capture the finer-grain modes accurately. Similar phenomena can be observed on the $CD_{naive}$ baseline, where the Euclidean assumption does not necessarily capture the curved Riemannian geometry.

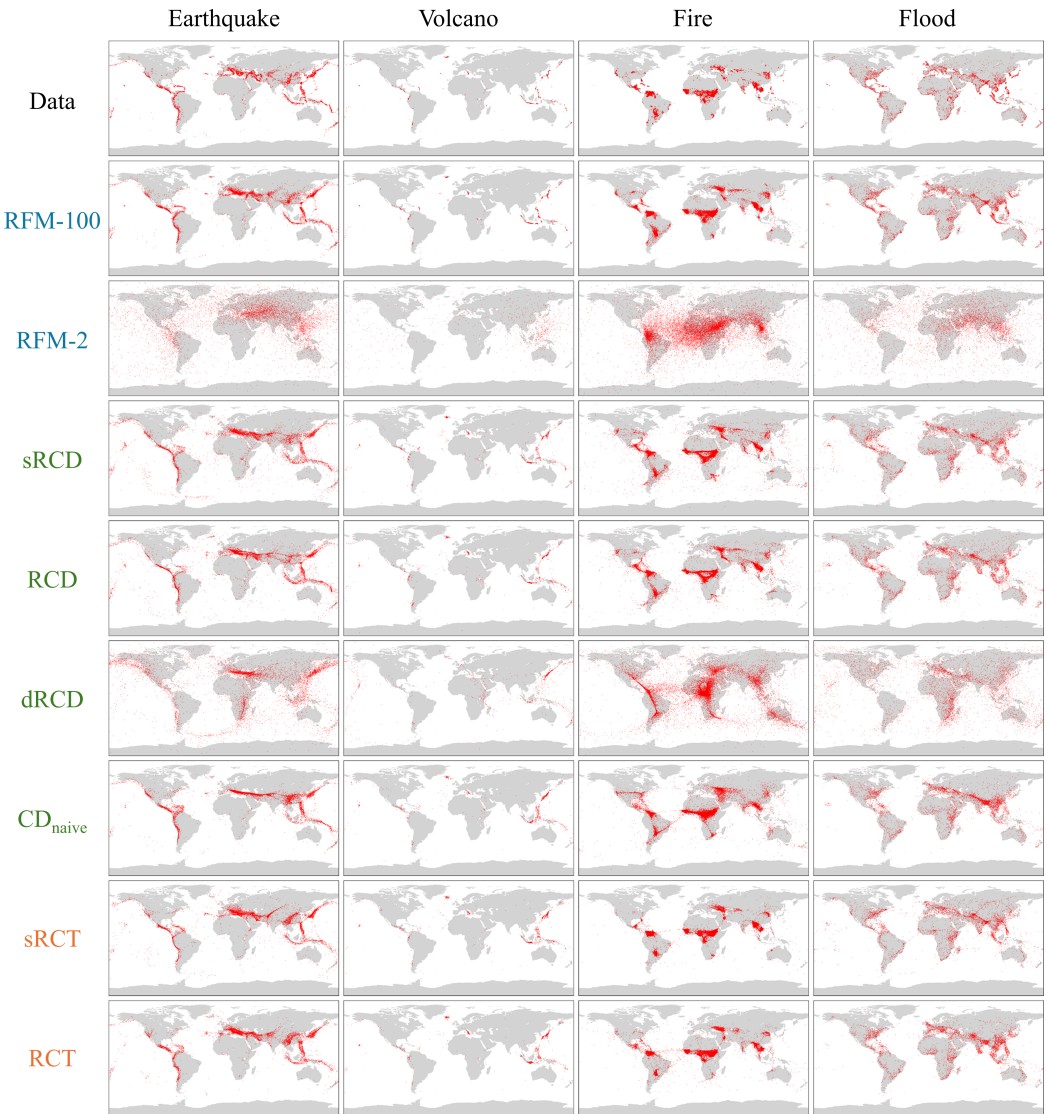

Figure 6: Generations on the four datasets on the 2-sphere. The FM, CD, and CT models are colored in blue, green, and orange.

