# OpenReview forum: "Riemannian Consistency Model"
_NeurIPS.cc/2025/Conference — NeurIPS 2025 poster_

### Official Review · Reviewer_J7YW · 2025-06-24

**Clarity:** 3
**Significance:** 2
**Originality:** 2
**Rating:** 5
**Confidence:** 4

**Summary:**

This work proposes a new way to train few-step generative models on Riemannian manifolds, taking inspiration from and generalising existing literature on consistency models. They show a simplified training objective that allows for simpler and faster computation, and, for small ($N = 2$), equal amounts of steps, they show that their method outperforms typical Riemannian flow matching.

**Questions:**

1. Do you have any theoretical results in terms of the learnt distribution? It seems that the authors have not delved too deep in Probability Theory over Riemannian manifolds—which is fine, but I think it would consolidate the paper theoretically rather substantially.
2. Have you tried any other datasets, such as those in [4]?
3. Have you tried to run [1] on the datasets you use?
4. To compare the different methods, it would have been particularly convenient to also obtain the NLL on the test sets. Is it possible to evaluate the NLL using your method?
5. I see no errors on your numbers. Have you re-run your experiments to obtain confidence intervals? They should be relatively cheap, considering the dimensionality of the considered datasets.

**Ethical Concerns:**

["NO or VERY MINOR ethics concerns only"]

**Final Justification:**

The authors have demonstrated through the rebuttals the soundness and novelty of their method, which does indeed set a new high in the field of few-step generative models over Riemannian manifolds. The discussions incurred will enhance the paper to match the standards required.

**Limitations:**

It is unclear whether the method could be scaled up to higher dimensions.

References:
---

[1] Peter Sorrenson, Felix Draxler, Armand Rousselot, Sander Hummerich, Ullrich Köthe (2023). Learning distributions on manifolds with free-form flows.

[2] Linqi Zhou, Stefano Ermon, Jiaming Song (2025). Inductive Moment Matching

[3] Dongjun Kim, Chieh-Hsin Lai, Wei-Hsiang Liao, Naoki Murata, Yuhta Takida, Toshimitsu Uesaka, Yutong He, Yuki Mitsufuji, Stefano Ermon (2023). Consistency Trajectory Models: Learning Probability Flow ODE Trajectory of Diffusion.

[4] Ricky T. Q. Chen, Yaron Lipman (2023). Flow matching on General Geometries.

**Paper Formatting Concerns:**

-

**Quality:**

2

**Strengths And Weaknesses:**

Strengths:
1. The paper presents a relatively simple and intuitive method to train a few-step generative model on a Riemannian manifold, which is a relatively unexplored topic.
2. The proposed method seems to improve significantly on its usual flow matching counterpart.
3. The authors illustrate with understandable examples their approach in this work, namely in section 3.3, and they present differential geometry in rather clear terms.

Weaknesses:

1. I believe that the need for few-step generative models on Riemannian geometries could have been better motivated in the introduction.
2. There are no theoretical guarantees relating to the learnt distribution of the model, although I do not believe it is of primary importance.
3. I find the argument at the end of section 3.1 (“The proof above immediately…”) particularly hand-wavy, and do not get how equation 8 is a natural consequence.
4. There is not much new conceptual content introduced in this work—which would have been fine if the experiments were more extensive.
5. The experiments are not extensive enough, as they show only low-dimensional cases (sphere, flat torus and $\mathrm{SO}(3)$).
6. The experiments include no comparable baselines, such as [1] (cited in section 5, which is perhaps the most important one to include), or Euclidean few-step generators [2, 3]. It is also quite unclear which consistency method has been used as a baseline for “Euclidean consistency models”: is it the authors’ framework on an Euclidean geometry, or another method, which only works on Euclidean geometry?

I believe that I would be happy to reconsider my assessment if given more empirical evidence: more datasets (in higher dimensions), more comparable baselines.

---

> ### Author Rebuttal · Authors · 2025-07-30
>
> We thank you for your comprehensive review and your high recognition of our proposed RCM's experimental improvement. We are happy to answer your questions as follows.
>
> ## Q1 Motivation
>
> Thanks for your suggestions. We will further elaborate on the motivations as follows. While diffusion or flow-based generative models that respect the intrinsic geometry have been proposed, one of their limitations is that they often **require hundreds to thousands of iterative sampling steps** to obtain decent results, which is a time-consuming process. While consistency models have been introduced to accelerate the Euclidean diffusion/flow models, **their generalization to Riemannian manifolds is non-trivial**. In this regard, we proposed RCM as a mathematically sound extension of CM to Riemannian manifolds, aiming to **enhance the inference time efficiency** of existing diffusion/flow models. In practice, such a few-step generation setup can accelerate important real-world applications, such as protein design (e.g., FrameFlow/FoldFlow, which operates on SE(3)), with significantly reduced inference time. Therefore, we hope RCM can have a profound impact on such AI4Science domains where Riemannian manifolds are encountered.
>
> ## Q2 Theoretical Guarantees of Convergence
>
> The theoretical guarantees of convergence can be established using Theorem 3.2 in our paper. We provide a brief sketch of proof as follows.
>
> - According to Theorem 3.2 in our paper, RCD and RCT are equivalent, assuming a perfectly learned pre-trained RFM model that approximates the marginal vector field.
> - According to Theorem 1 in the CM paper, a perfect CD loss indicates a perfect recovery of the marginal probability paths. Such a result can also be extended to Riemannian cases with some regularity conditions.
> - According to the RFM paper, Theorem 3.1, a perfect conditional Riemannian flow matching model can perfectly approximate the marginal vector field, thus giving the true marginal probability path.
>
> Combining the three results above, we arrive at the conclusion that, given a perfect pre-trained RFM, both RCD and RCT can recover the ground truth data distribution if perfectly optimized.
>
> Despite theoretical guarantees, the training procedure in practice cannot be perfect due to limited data, resources, and numerical issues. Still, our empirical evaluation demonstrates the effectiveness of RCM in practice in such a few-step generation setup.
>
> ## Q3 Alternative Loss in Eq.8
>
> We apologize for the confusion and provide a detailed mathematical deduction below. Specifically, for the loss value, we have
> $$\langle f_{\theta^-}-f_\theta+\dot f_{\theta^-},\dot f_{\theta^-}\rangle_g=\\|\dot f_{\theta^-}\\|\_g^2.$$
> For the loss gradient, we have
> $$
> \nabla_\theta\langle f_{\theta^-}-f_\theta+\dot f_{\theta^-},\dot f_{\theta^-}\rangle_g
> =\nabla_\theta\langle f_{\theta^-}-f_\theta,\dot f_{\theta^-}\rangle_g
> =\langle \nabla_\theta (f_{\theta^-}-f_\theta),\dot f_{\theta^-}\rangle_g\\
> =-\langle \nabla_\theta f_\theta,\dot f_{\theta^-}\rangle_g\\
> \approx\frac{\Delta t}{2}\nabla_\theta d_g^2(f_\theta,\tilde{f}_{\theta^-})
> $$
> where the last approximation follows the deduction in Eq.7. This is also the technique used in the sCM paper.
>
> ## Q4 Conceptual Innovation
>
> We respectfully disagree with the reviewer's argument that our work has "not much new conceptual content". Our theoretical contributions and innovations on the few-step generation setup on the Riemannian manifold have been unanimously acknowledged by other reviewers. We further elaborate on our new contributions as follows.
>
> - Regarding the task, our work is the **first to systematically explore few-step generative modeling** for Riemanian diffusion/flow-based models. Although generative frameworks on Riemannian manifolds exist, they predominantly require a large number of steps to achieve decent results.
> - To the best of our knowledge, our RCM is the **first to introduce covariant derivatives and differentials of the exponential map** in the context of generative modeling. We provide a **novel kinematics perspective** in Section 3.3 that links these important mathematical concepts with efficient generative models on a Riemannian manifold. RCM not only fills this gap but also provides solid theoretical grounding and substantial empirical evidence, further supporting our claim that such Riemannian information is crucial for generating good results.
> - Our framework is **theoretically grounded**, with Theorem 3.2 guaranteeing the equivalence between RCD and RCT. Due to the complexity of Riemannian geometries, such a result is a **non-trivial extension** of the Euclidean counterparts.
>
> ## Q5 Scalability to Higher Dimension
>
> We provide additional scalability experiments on the manifold with up to 256 dimensions. Please kindly refer to our rebuttal to Reviewer aiD6, Q1. To summarize, our proposed **RCM consistently outperformed the few-step RFM and the Euclidean CM** (CD-naive). Such experimental results also necessitate the need for Riemannian constraints, as Euclidean CM often fails to capture the manifold constraint, especially when the dimension is large (e.g., periodicity in our scalability experiments).
>
> ## Q6 Additional Baselines
> We first note that our experiments have already included the CD-naive baseline that corresponds to the standard CM model for Euclidean data. Such a CM model has proven powerful and effective in few-step image generation and thus offers a strong baseline for comparison.
>
> Nonetheless, we follow your suggestion to conduct additional experiments using the FFF model on our datasets. The results are summarized in the table below:
>
> | Dataset | Earthquake | Volcano | Fire | Flood | Protein | RNA | Fisher |
> |---|:---:|:---:|:---:|:---:|:---:|:---:|:---:|
> | Metric | 2.76 KLD | 22.57 KLD | 2.02 KLD | 2.34 KLD | 4.96×10⁻² MMD  | 8.52×10⁻² MMD  | 6.67×10⁻² MMD  |
>
> It can be seen that FFF provides decent generation results for larger datasets, but **still falls short compared to the best RCM variant** on all datasets. Interestingly, for smaller datasets like Volcano and RNA, FFF's performance is considerably worse, although it still achieves better performance compared to the few-step RFM setup. Such additional experimental results further demonstrate the effectiveness of RCM.
>
> ## Q7 NLL Calculation
>
> We fully understand that NLL is one of the important metrics for generative models. However, we noted that, due to the few-step generation setup, NLL cannot be calculated as the original RFM model. We further elaborate on the reasons as follows.
>
> We first note that we can consider diffusion/flow models as a pushforward transformation of the prior probability distribution. The change-of-measure (change-of-probability) terms can be calculated as
> $$\nabla\_\theta\log|f'\_\theta(x)|=\mathrm{tr}(\nabla\_\theta f'\_\theta(x)f^{'-1}\_\theta(z)),$$
> where $z=f\_\theta(x)$ (Theorem 1 in the FFF paper). Therefore,
>
> - Normalizing flow can naturally compute this quantity as its model is designed such that the trace of the Jacobian of each layer in the model can be calculated in closed form.
> - FFF approaches this by learning an inverse model $g_\phi$ such that $g_\phi(f_\theta(x))\equiv x$. In this way, the inverse Jacobian can be approximated using the learned $g$.
> - The standard flow matching model, as a special case of continuous normalizing flow, also enjoys exact likelihood calculation by integrating the model Jacobian. Here, we note that the continuous-time is essential, as the flow is always invertible locally. In this way, the inverse is simply the negative vector field. However, for CM models that shortcut the probability path, such an invertibility no longer holds. In other words, we do not know the inverse function of $f_\theta$ for NLL calculation. It is, though, possible to combine FFF to CM models to learn an "inverse" shortcut flow for NLL estimation.
>
> Therefore, we resorted to alternative metrics for a fair comparison between baselines. The KDE with KLD approach follows the Riemannian normalizing flow paper. MMD is also widely used in point cloud matching. In this regard, our existing metrics already provide comprehensive evaluations that demonstrate the effectiveness of RCM over the baselines.
>
> ## Q8 Error bars
>
> We thank you for your suggestions on adding error bars to the evaluation metrics. In the table below, we provide the new results with three repeats on the Earthquake dataset as an example. It can be seen that most original metrics fall inside the new variance range. Interestingly, all RCT models have higher variance than the CD models, probably due to the noisier training procedure with the teacher model. Still, given the current results, **our improvements over the baselines are indeed significant**. Due to the limits of time and rebuttal space, we will add other results in the revised manuscript.
>
> | Model | RFM-100 | RFM-2 | sRCD | RCD | dRCD | CD-naive | sRCT | RCT |
> |---|:---:|:---:|:---:|:---:|:---:|:---:|:---:|:---:|
> | KLD | 1.57 ± 0.26 | 10.57 ± 0.50 | **1.97 ± 0.06** | 2.12 ± 0.04 | 6.38 ± 0.06 | 3.14 ± 0.02 | 2.27 ± 0.40 | 2.22 ± 0.23 |

---

> > ### Comment · Reviewer_J7YW · 2025-08-04
> >
> > I would like to thank the authors for the thorough answer. It has allayed most of my concerns. I have just a few questions remaining, but I shall upgrade my score to 5 already:
> >
> > 1. I understand the issue with Normalizing Flows, and other architectures to get Jacobians or approximations thereof. I just wonder if there is a way to still compute the likelihood in a relatively simple way. Is it perhaps possible to retrieve the instantaneous vector field, for instance, and then to integrate it as usual? The likelihood would be slow to compute, but it could be potentially some source of confirmation for the learnt consistency model.
> >
> > 2. I see that the dataset that incurs the most difference in performance between FFF and your method is "Volcano". Any intuition on why this is the case for this dataset specifically? Perhaps more sparsity than the others? Could better training close the gap?
> >
> > Thank you again for your answers.

---

> > > ### Author Response · Authors · 2025-08-04
> > >
> > > We sincerely thank you for raising your score as recognition of our work, and we are glad that our response and results have addressed your concerns. We are more than willing to engage in further discussions with you on the theoretical parts of RCM.
> > >
> > > ## NLL Calculation.
> > > As we have mentioned in our previous rebuttal, due to the few-step setup, **retrieving the reverse instantaneous vector field is non-trivial**. As a concrete example, consider the two-step generation setup in which we go from 0 (noise) to 0.5 to 1 (prediction). While the vector field at t=0.5 does send the data from 0.5 to 1, it is not guaranteed that the negation of the vector field will send the data from 0.5 to 0. Therefore, the instantaneous vector field is not a simple negation of the predicted vector field, unlike in the continuous flow matching.
> > >
> > > Inspired by this observation, we propose one potential approach that enables NLL calculation in our setup: we may **learn a reverse consistency model that shortcuts the reverse probability path** from t=1 (data) to 0 (noise). In this way, we can use the Jacobian of such a reverse model for the likelihood calculation. As you may have noticed, such an approach shared a similar spirit to FFF, learning the reverse process to eliminate the need for closed-form Jacobians. This echoes our previous rebuttal that RCM and FFF are more orthogonal approaches that could be combined together. We sincerely thank your insight into FFF, and we will keep exploring this direction in our future work.
> > >
> > > ## Performance Analysis on FFF
> > > Regarding the worse performance on the Volcano dataset, we fully agree with your opinion and hypothesize that it was because of the data sparsity. Volcano only has 829 data points, which is 8-15 times smaller than the other datasets. After looking into the generation results, we noticed that the FFF generation is more dispersed than the CM-based model, probably because it is harder to push forward the prior uniform noise distribution into a highly concentrated data distribution. It is possible that more training will improve performance slightly. Nonetheless, as we followed the original hyperparameters in the FFF repo, we believe current results already offer a fair comparison.
> > >
> > > We deeply appreciate your insightful feedback and kind engagement throughout this review process. We are committed to further polishing our work, and we will add the new results to the revised manuscript to make our work more comprehensive.

---

> > > > ### Comment · Reviewer_J7YW · 2025-08-06
> > > >
> > > > Thank you very much for your further comments. You have addressed all my concerns.

---

### Official Review · Reviewer_iNgD · 2025-07-02

**Clarity:** 4
**Significance:** 2
**Originality:** 3
**Rating:** 4
**Confidence:** 3

**Summary:**

The paper introduces Riemannian Consistency Models (RCM) by extending the traditional consistency model to Riemannian manifolds. In the process, they derive a closed-form continuous-time objective tailored to manifold-specific operations. By applying the marginalization trick, the authors prove the mathematical equivalence of Riemannian consistency training and distillation. Additionally, a simpler objective is proposed that avoids computing differentials of the exponential map, yet maintains performance, with a kinematic interpretation offering intuitive insight into the method.

**Questions:**

- _[Line 86-87]_ Kindly define the weights $w_t$. Do they follow a similar specification to that in [1] or [3]?
- _[Line 127]_ As pertaining will always include an inherent additional error due to generalization to domains, what should be a sufficient condition for $\dot{x}$ such that RCM converges to the target distribution?
- _[Line 245]_ Is there any rationale behind sticking to the linear scheduler?
- _[Line 252]_ The choice of KDE to obtain density estimates is questionable, since in higher dimensions (even as low as 10-15), choosing suitable bandwidths becomes difficult. At best, carefully chosen kernels can recover error bounds adapted to intrinsic dimensions under the manifold hypothesis (see [4]), theoretically. In practice, there is hardly any improvement.
- There remains the issue of training instability that can occur when consistency models are trained from scratch in a naïve manner—specifically, by directly optimizing the objective as in (4) while closely adhering to $\Delta t \approx 0$. The authors of [3] call it the Curse of Consistency. In particular, the model tends to converge slowly after a certain point, perhaps due to accumulated consistency errors from each interval. Does the proposed model experience anything similar, since it follows a simple training regime akin to [2] (following _Line 87_,  $\Delta t$ is quite small for large $N$)?
- Heuristically speaking, deep generative models often can be observed to work well enough, being agnostic to the fact that underlying data distributions are supported on intrinsic structures embedded into the ambient space. They silently adopt the intrinsic dimensions in their error rates (e.g., GANs [5], WAE [6], etc. However, to my knowledge, error bounds on diffusion model-based generations are yet to be shown to follow the same.). In this context, does RCM really generalize the Euclidean case _[Line 139]_ or rather extend the framework onto manifolds? Given the additional costs due to hyperbolic/Riemannian operations, is it really meaningful to propose a new model altogether?
- Also, kindly respond to the points raised as 'Weakness'.

> [3] Zhengyang Geng, Ashwini Pokle, Weijian Luo, Justin Lin, and J Zico Kolter. 2025. Consistency Models Made Easy. In The Thirteenth International Conference on Learning Representations.
>
> [4] Jisu Kim, Jaehyeok Shin, Alessandro Rinaldo, and Larry Wasserman. Uniform convergence rate of the kernel density estimator adaptive to intrinsic volume dimension. In International Conference on Machine Learning, pp. 3398-3407. PMLR, 2019.
>
> [5] Jian Huang, Yuling Jiao, Zhen Li, Shiao Liu, Yang Wang, and Yunfei Yang. An error analysis of generative adversarial networks for learning distributions. Journal of machine learning research 23, no. 116 (2022): 1-43.
>
> [6] Saptarshi Chakraborty and Peter Bartlett. A Statistical Analysis of Wasserstein Autoencoders for Intrinsically Low-dimensional Data. In The Twelfth International Conference on Learning Representations. 2024.

**Ethical Concerns:**

["NO or VERY MINOR ethics concerns only"]

**Final Justification:**

The authors' responses have been prompt and insightful. Considering all the reviews, the subsequent discussions, and my initial assessment, I believe it is appropriate to revise my score upward.

**Limitations:**

While the article does not have a dedicated section explaining limitations, the apparent ones mostly relate to empirical validation.

**Paper Formatting Concerns:**

The are no noticeable formatting issues.

**Quality:**

3

**Strengths And Weaknesses:**

**Strength**
> The paper is very well-written, easy to follow, with sufficient discussion on similar approaches extending diffusion/flow models in non-Euclidean data supports. Supplementary notes on rudimentary notions of geometry are appreciable.

**Weakness**
>- While the motivation to adopt consistency models, amenable to data supported on a Riemannian manifold, is clear, the paper lacks empirical validation. For example, what is the extent of improvement in inference time (or convergence) over both Riemannian diffusion models [1] and/or vanilla consistency models [2].
>- Given that the calculations in a non-Euclidean space are far more computationally demanding compared to their Euclidean counterparts, the paper must show that the advantages of adopting this approach outweigh the added processing cost. This is missing in the current article.
>- Computational infrastructure under which experiments were carried out, code repositories, and specific execution details seem to be missing in the current article.


> [1] Huang, Chin-Wei, Milad Aghajohari, Joey Bose, Prakash Panangaden, and Aaron C. Courville. Riemannian diffusion models. Advances in Neural Information Processing Systems 35 (2022): 2750-2761
>
> [2] Yang Song, Prafulla Dhariwal, Mark Chen, and Ilya Sutskever. 2023. Consistency models. In Proceedings of the 40th International Conference on Machine Learning (ICML'23), Vol. 202. JMLR.org, Article 1335, 32211–32252.

---

> ### Author Rebuttal · Authors · 2025-07-30
>
> We sincerely appreciate your comprehensive review and your recognition of the clear structure of our work. We are happy to address any remaining questions you may have.
>
> ## Q1 Empirical Improvement over Existing Work
>
> We first thank your recognition of our theoretical contributions and applications. Here, we provide a more comprehensive discussion on the empirical improvements.
>
> The inference time improvement of RCM, as is the case for regular CMs for Euclidean data, is straightforward. This is because the inference code is identical to that in the RFM except for a smaller number of Euler steps. As a concrete example, in most of our experimental setups, the RFM model requires approximately 100 inference steps. In contrast, our RCM runs with just two steps, resulting in a **50x speedup during inference**. Furthermore, compared to RFM or Riemannian diffusion under a few-step setup, our results also demonstrate **better generation quality** (as shown in the RFM-2 column), highlighting the effectiveness of RCM in this context on Riemannian manifolds. Lastly, we provided the CD-naive baseline, which is **equivalent to the classic consistency model on Euclidean data**. On most datasets, even after the final projection to the manifold, the Euclidean assumptions often fail to capture the true data distribution, resulting in lower generation quality and necessitating manifold constraints. In short, our experimental results demonstrate the effectiveness of RCM in terms of both **inference efficiency and few-step generation quality**, as well as the necessity of the Riemannian constraint, which motivates our framework.
>
> ## Q2 Computational Cost & Advantage of Riemannian Constraint
>
> We first point out what might be a **misconception** that the additional Riemannian operation is "computationally demanding". In fact, we provide a comprehensive analysis of the overhead, both asymptotic and empirical, compared to the Euclidean CM model. Please kindly refer to Reviewer nYt4, Q5, for details. To summarize, such an overhead is **asymptotically negligible** as model size increases (as it is model-independent) and **practically small** (within 10% even for a complex manifold of SO(3) and a small model). We also provide additional scalability experiments in our rebuttal to Reviewer aiD6, Q1, in which our proposed **RCM consistently outperformed the few-step RFM and the Euclidean CM** (CD-naive) on a high-dimensional manifold (up to 256 dim). Such experimental evidence validates the need for our proposed Riemannian corrections, as Euclidean CM often fails to capture the manifold constraint, especially when the dimension is large (e.g., periodicity in our scalability experiments).
>
> ## Q3 Computational Infrastructure
>
> We have provided the detailed experimental setup in Appendix C (including the techniques we used to stabilize the training). Here, we provide additional infrastructure specifications. All experiments were carried out on a single A100. The maximum GPU memory consumed is only around 5GB, which can be easily fit into GPUs with smaller memory. As we fixed the number of iterations, the whole training of each variant of CM models took about 4 hours regardless of the dataset size.
>
> For code, we will publicly release our code in an effort to inspire future work once our paper is published.
>
> ## Q4 Theoretical Guarantees on Convergence
> We first note that RCT formulation, as guaranteed by our Theorem 3.2, does not rely on pre-trained models and will converge to the data distribution assuming a perfectly trained model.
>
> For RCD that does rely on a pre-trained model, the asymptotic analysis in the CM paper, Theorem 1, can be adapted for RCM under some regularity conditions. Informally speaking, such a theorem guarantees that if a perfect CD model is trained (with the pre-trained flow-based model), it can perfectly "shortcut" the marginal probability path. Therefore, theoretically, the errors can be fully bounded by the errors of the pre-trained flow-based model.
>
> In practice, we observe that the CM model is generally more challenging to train than the flow matching model. See our discussion on the training instability in Q6. In this way, extending CM to Riemannian manifolds is a non-trivial task.
>
> ## Q5 Impact of KDE Bandwidth
>
> We appreciate your insightful suggestions regarding the impact of KDE bandwidth. We chose the bandwidth parameters following [1]. Additionally, we provide additional ablations on the impact of the KDE bandwidth in the table below. It can be seen that, although the choice of bandwidth did impact the numbers of KLD, their relative rankings between different models remain largely unchanged. Therefore, we believe it remains an informative metric for measuring generation quality. As a more concrete example, we run additional evaluation on the Earthquake dataset with a KDE bandwidth of 0.04 (instead of 0.02). The results are as follows. Although the specific KLD values do change a lot, the relative ordering is perfectly preserved.
>
> | Model | RFM-100 | RFM-2 | sRCD | RCD | dRCD | CD-naive | sRCT | RCT |
> |---|:---:|:---:|:---:|:---:|:---:|:---:|:---:|:---:|
> | KLD | 0.40 | 6.13 | **0.66** | 0.75 | 3.77 | 1.48 | 1.04 | 0.94 |
>
> [1] Mathieu, Emile, and Maximilian Nickel. "Riemannian continuous normalizing flows." Advances in neural information processing systems 33 (2020): 2503-2515.
>
> ## Q6 Training Instability
> It is already known that training CMs will generally encounter more instability (e.g., in the sCM paper).
> Similar to the original CM, we also encountered instabilities during training, both for discrete-time RCM that uses Δ𝑡 and for continuous-time RCM that relies on the model JVP. For the discrete-time RCM (referred to as dRCD in the results), we follow the original CM paper and use $\Delta t=0.01$. We observed NaN loss sometimes for $\Delta t=1𝑒-3$ and almost surely NaN for $\Delta t=1𝑒-4$. The results echo with the CM and  ECT paper, where the former also uses a $\Delta t$ around 0.01.
>
> For the continuous-time scenario, we use multiple techniques inspired by sCM to mitigate the instability caused by the initially noisy JVP signals. The details were provided in Appendix C.2, and additional ablations are provided in our rebuttal to Reviewer nYt4, Q1.
>
> ## Q7 Miscellaneous Questions
>
> - **Weight** $w_t$. Yes, we follow the CM and sCM paper in choosing the weight as $t^2/(1-t)^2$. In practice, we do not experience a significant impact from the choice of scheduler, as they perform equally well.
> - **Scheduler**. We simply follow RFM to use the linear scheduler. Despite fixing this so far, our theoretical framework allows for the flexibility of other schedulers.
> - **Comparison to other generative models (GAN, VAE)**. We interpret your question as the advantage of enforcing the manifold constraint over the pure data-driven approaches (with GAN/VAE) that ignore such a constraint. While we agree that the bounds for VAE/GAN are true, they only hold asymptotically with an infinite amount of data with perfect learning assumptions. In practice, they do not necessarily hold, as demonstrated in our comparisons with the CD-naive baseline and discussions in Q2. Therefore, RCM provides a **more data-efficient approach** in which the **manifold constraint holds by design**. Regarding the computational cost, we have clarified in Q2 that the additional Riemannian operations add negligible overhead.

---

> > ### Comment · Reviewer_iNgD · 2025-08-04
> >
> > I appreciate the rigorous responses by the authors.
> >
> > * [_Speedup during inference_] Could the authors be a bit more elaborate on the speedup? Does a ~50x improvement occur across all experiments? Is it expected to vary significantly based on the target data distribution? Given that the proposed framework potentially generalizes the 'Euclidean' CM, does such a speedup hold up for 'Euclidean data' as well (e.g., images, as in the case of CMs; I say this despite acknowledging the manifold hypothesis)? Any observation, if there, from experience would be helpful (and sufficient) for the last question, as it would be overburdening to ask for experimental evidence at this stage of rebuttals (I ask this acknowledging the authors' response to Reviewer nYt4, Q5).
> >
> > * [_Impact of KDE Bandwidth_] Regarding the concern related to the usage of KDE, the results provided in response to Reviewer aiD6, Q1 are rather reassuring. While I still remain opposed to prescribing KDE naively, especially when dimensions are as high as 256, can the authors explain why the degradation is much more pronounced in e.g., CT naive-2, compared to RCT?

---

> > > ### Author Response · Authors · 2025-08-04
> > >
> > > We sincerely appreciate your kind engagement in the discussion. We are more than happy to provide further details and clarifications on your additional questions.
> > >
> > > ## Speedup Inference
> > > We'd like to first clarify that our response to nYt4 Q5 details the small overhead introduced during **training**, as it is the only stage where most of the computation-heavy operations (covariant derivative, differentials of exp map) happen. Once the model is trained, its **inference cost per time-step would be exactly the same** as the original flow matching model. Therefore, with RCM's fewer steps (2-step) of generation than the RFM model (100-step), our 50x inference speedup is **consistent for all CMs and independent of the data distribution**. More concretely, we outline the computations that happen in each inference step for RFM and RCM as follows:
> > >
> > > 1. Predict the current vector field using the learned model $v_\theta(x_t,t)$. This step is **exactly the same** for RFM and RCM, and is also the bottleneck in computation cost.
> > > 2. Proceed 1 step forward to $x_{t'}$ where $t'>t$ is the next timestep in inference trajectory. For RFM, it's $\exp_{x_t}(t'-t)v_\theta$. For RCM, it's $\exp_{x_0}(t'\log_{x_0}(\exp_{x_t}(1-t)v_\theta))$ (following the original CM paper). The computational cost for this part is negligible.
> > >
> > > To provide more concrete quantitative results, the actual **inference speedup** of RCD-2 compared to RFM-100 is provided in the table below. The results are as expected, with speedups slightly below 50.
> > >
> > > | Dataset | Earth | RNA | SO(3) |
> > > |---|---|---|---|
> > > | Speedup | x46.5 | x48.0 | x43.2 |
> > >
> > > Regarding the generalization to Euclidean data, our RCM will exactly reduce to the (continuous-time) CM models when instantiated on the Euclidean manifold. Therefore, as also demonstrated in the original CM paper, such a speedup also holds, proportional to the ratio of the inference steps required between the CM and diffusion/flow matching sampling.
> > >
> > > ## KDE Bandwidth & Evaluation Metric
> > > We are glad that our ablations on the KDE bandwidth and results on high-dimensional manifold have addressed your concerns. Regarding the experiments on the high-dimensional tori, we first noted that the **KDE was not used as the metric for this task**. Instead, we estimated the Gaussian parameters using MLE and calculated the **Fréchet distance** (Wasserstein-2 distance) to the ground truth standard normal parameters (see our rebuttal to Reviewer aiD6, Q1).
> > >
> > > Regarding the worse performance of CD-naive (Euclidean CD) on higher dimensions, we hypothesize it is because the difficulty in maintaining the manifold constraint grows exponentially as the dimension of the manifold grows. Specifically, the wrapped Gaussian dataset in our scalability experiments is centered at the origin, meaning that each of the $2^n$ corners of the hypercube has a non-trivial density that the model should capture and maintain the periodic boundary conditions. Therefore, for the CD-naive model, enforcing as a periodic condition is extremely hard given the high data dimension, as the model has to **coordinate globally at each corner**. In contrast, for RCM, such a periodic condition is **guaranteed by design** as the generations are properly wrapped onto the manifold. Such a result further demonstrates our claim that the manifold constraint is crucial for generative modeling.
> > >
> > > Once again, we thank your insightful questions. Should you have any further questions, we are more than happy to engage in additional discussions with you.

---

### Official Review · Reviewer_aiD6 · 2025-07-03

**Clarity:** 2
**Significance:** 3
**Originality:** 2
**Rating:** 4
**Confidence:** 3

**Summary:**

This paper proposes the Riemannian Consistency Model (RCM), allowing few-step generative modeling on Riemannian manifolds (e.g., spheres, tori, SO(3)). It adapts consistency models to respect manifold geometry using exponential maps and covariant derivatives, and includes both theoretical and practical improvements. RCM outperforms previous Euclidean and flow matching baselines in experiments.

**Questions:**

Please see the weaknesses.

**Ethical Concerns:**

["NO or VERY MINOR ethics concerns only"]

**Final Justification:**

Authors have addressed my concerns.

**Limitations:**

Yes, although authors mention experiments on larger datasets are their limitations, however, baseline [1] do report performance on these datasets. I would advise authors to include at least one such dataset to showcase the scalability and others can fall under this limitation.

**Paper Formatting Concerns:**

Typo in Eq 2 --> missing the $\theta^-$

**Quality:**

3

**Strengths And Weaknesses:**

Strengths:
- Extends consistency models to non-Euclidean manifolds and provides the simplified training objective
- Strong theoretical analysis and proofs to support the proposed approach
- Results on multiple manifolds and datasets with respect to the selected baselines

Weaknesses:
- The main experimental limitations of the RCM paper, compared to RFM, are a narrower focus on simple manifolds, lack of high-dimensional and general (mesh/bounded) manifold experiments after the literature [1].
- Missing baselines: Diffusion Model, Riemannian Diffusion Model, Riemannian Score-Model, etc.
- Missing ablation: (NFE vs. Performance) plot could give a better overview of the effectiveness of the proposed approach.

[1] Chen, Ricky TQ, and Yaron Lipman. "Riemannian flow matching on general geometries." arXiv e-prints (2023): arXiv-2302.

---

> ### Author Rebuttal · Authors · 2025-07-30
>
> We sincerely thank you for recognizing our work's strong analysis and proof to extend consistency models to multiple Riemannian manifolds. We are happy to answer your questions as follows.
>
> ## Q1 Application to High-Dimensional Manifolds
>
> We thank you for your suggestions on the additional experiments to demonstrate the scalability of our RCM to high-dimensional manifolds. To achieve this, we follow the RFM and [1] to conduct additional experiments on high-dimensional synthetic data on the $n$-torus. Following [1], we used the truncated standard Gaussian on the domain $[0,2\pi]^n$ as the target and sampled 10k points as the training set. Each model was trained for 2000 epochs, and we estimated the Gaussian parameters using maximum likelihood estimation with truncated Gaussians on 10k generated samples. We then calculated the Fréchet distance (Wasserstein-2 distance) to the ground truth standard normal parameters as the metric. For a manifold dimension of $2^k$ where $k=1,2,...,8$, the results are summarized in the following table.
>
> | dim | 2 | 4 | 8 | 16 | 32 | 64 | 128 | 256 |
> |---|:---:|:---:|:---:|:---:|:---:|:---:|:---:|:---:|
> | RFM-2 | 0.52 | 0.70 | 1.01 | 1.41 | 1.95 | 1.47 | 1.83 | 2.41 |
> | RCT-2 | **0.22** | **0.31** | **0.54** | **0.81** | **0.46** | **0.58** | **0.62** | **0.96** |
> | CT naive-2 | 0.73 | 2.69 | 1.58 | 2.16 | 2.41 | 24.80 | 35.16 | 49.24 |
>
> It can be clearly seen that our proposed RCM model **consistently outperforms the RFM model in a few-step generation setup**. Interestingly, the naive Euclidean CT baseline quickly fails as the manifold growing dimension adds to the difficulty in enforcing the periodic constraint in every dimension.
>
> We also provide an asymptotical analysis on the additional cost of Riemannian operations (e.g., covariant derivative). Please kindly refer to our rebuttal to Reviewer nYt4, Q5. In short, the overheads regarding the additional Riemannian operators are **small compared to the naive Euclidean CM setup and are also model-independent**. Therefore, we believe our proposed RCM can scale to large data and models.
>
> Regarding the extension to general manifolds like meshes, we first noted that, in principle, like Riemannian flow matching, RCM can be used for general geometries but requires expensive simulation (of the geodesic and covariant derivatives). Therefore, we acknowledge that this is one of the limitations of our model so far, which we are actively exploring. We noted that most existing work, like [1] and [2], operates on the manifolds with known exponential and logarithm maps. We emphasize our theoretical contributions in such manifolds, which already exhibit profound practical applications as indicated in previous work.
>
> [1] De Bortoli, Valentin, et al. "Riemannian score-based generative modelling." Advances in neural information processing systems 35 (2022): 2406-2422.
>
> [2] Mathieu, Emile, and Maximilian Nickel. "Riemannian continuous normalizing flows." Advances in neural information processing systems 33 (2020): 2503-2515.
>
>
> ## Q2 Baselines Question
>
> Comprehensive baselines have always been a key target for us. However, we noted that the Riemannian diffusion and Riemannian score model are not directly comparable baselines in our setup, based on the following reasons:
>
> - Our proposed RCM framework **targets the few-step generation setup** on the Riemannian manifold, whereas the above two baselines require 100-1000 steps for decent results. They are baselines to models like RFM but **not directly comparable** to our few-step generation results.
> - Even for simple geometries like spheres, the diffusion-based approach may require **expensive simulation** of the model Jacobian trace (as detailed in the RFM paper, Appendix D), due to its reliance on the Gaussian-like noise.
> - The Riemannian diffusion and score models fall short of the RFM model (in the RFM paper results).
>
> Instead, our baselines comprise comparable algorithms in the few-step generation setup on the Riemannian manifold, e.g., the original CM model, designed for Euclidean data (referred to as CD/CT-naive in our paper), and the RFM with its few-step generation results. In this way, our baselines already covered the comparable algorithms and demonstrated the effectiveness of our proposed RCM.
>
> In addition, we provide additional baseline results of FFF, a Riemannian normalizing flow that enables one-step generation, as per request by Reviewer J7YW, Q6. RCM consistently outperforms this new baseline, further demonstrating its effectiveness.
>
> ## Q3 Performance vs NFEs
>
> We thank your insightful suggestions on further ablations of the performance vs NFEs. Below, we provide comprehensive results on the Earthquake dataset, examining the impact of NFEs on generation quality (measured by KDE-KLD in this case). The best results, except for RFM, are highlighted in bold, while the second-best result is italicized.
>
> | NFE | RFM | sRCD | RCD | dRCD | CD-naive | sRCT | RCT |
> |---|:---:|:---:|:---:|:---:|:---:|:---:|:---:|
> | 2 | 10.31 | **1.91** | _2.14_ | 6.52 | 3.14 | 2.26 | 2.40 |
> | 5 | 4.74 | _1.83_ | **1.76** | 4.10 | 2.78 | 1.84 | 2.64 |
> | 10 | 3.24 | _2.09_ | **1.82** | 3.87 | 3.02 | 2.31 | 2.72 |
> | 20 | 1.90 | _2.32_ | **2.10** | 4.06 | 2.96 | 2.30 | 2.66 |
> | 50 | 1.53 | _2.41_ | **2.04** | 4.18 | 3.12 | 2.23 | 2.78 |
> | 100 | 1.47 | 2.64 | **2.02** | 4.72 | 3.16 | _2.15_ | 2.75 |
>
> We have the following interesting observations:
>
> - For the RFM model, increasing the NFEs has a significant impact on the generation quality, as its learned marginal vector fields are not necessarily straight for few-step generation.
> - On the other hand, RCD & RCT generations are more stable with respect to the NFEs, achieving consistently good generation quality. This indicates RCM can indeed shortcut the probability path on the Riemannian manifold.
> - RCM variants **consistently outperform the baselines**, especially the naive Euclidean CM approach, demonstrating the necessity of Riemannian constraints. For RFM, the two lines intersect at around 20 NFEs, below which the RFM generation quality drastically drops.
>
> It shall be noted that the speed-quality tradeoff for the standard flow matching model does not apply to CMs. Our NFE vs performance results generally echo the findings in the consistency trajectory model [3], in which high NFEs may instead lead to poorer generations.
>
> We also observe similar trends in other manifolds where RCD & RCT models have stable generation quality compared to RFM. Due to the limited space, we will add these results to our revised manuscript.
>
>
> [3] Kim, Dongjun, et al. "Consistency trajectory models: Learning probability flow ode trajectory of diffusion." arXiv preprint arXiv:2310.02279 (2023).

---

> > ### Comment · Reviewer_aiD6 · 2025-08-04
> >
> > I thank the authors for providing the detailed response. This somewhat improves my understanding of the work and happy to improve the avg score.
> >
> > However, I would like to note that I'm not an expert in these downstream tasks. Hence, would like to keep my confidence low.

---

> > > ### Author Response · Authors · 2025-08-04
> > >
> > > We sincerely thank your positive feedback, and we are glad that our response and additional results have successfully addressed your questions. We appreciate your insightful reviews in helping to make our work more comprehensive and solid. We are committed to further polishing our RCM and will include the additional results in our revised manuscript for a larger audience.

---

### Official Review · Reviewer_nYt4 · 2025-07-03

**Clarity:** 3
**Significance:** 3
**Originality:** 3
**Rating:** 5
**Confidence:** 2

**Summary:**

This paper introduces the Riemannian Consistency Model (RCM), a framework to extend consistency models to general Riemannian manifolds for fast, few-step generation. The authors present a principled approach using the exponential map and covariant derivative to correctly enforce the consistency objective while respecting the manifold's geometry. The work derives closed-form objectives for both distillation-based (RCD) and training-from-scratch (RCT) versions of the model and proves their theoretical equivalence. A key practical contribution is a simplified objective that is easier to implement without sacrificing performance. Through extensive experiments on spheres, tori, and the SO(3) rotation group, the paper demonstrates that RCM significantly outperforms naive baselines and few-step flow matching, highlighting the importance of a geometry-aware approach for generative modeling on non-Euclidean data.

**Questions:**

- Your appendix mentions the use of "Tangent Warmup" and "Tangent Clipping" for stable training. Could you reflect on the practical difficulty of working with the proposed objectives? What is the primary source of the instability that necessitates these heuristics, and does the simplified sRCM objective require them to the same extent as the full RCM?
- The paper successfully applies RCM to tori, spheres, and SO(3). For which other classes of manifolds are the required geometric quantities (exponential/logarithm maps, Christoffel symbols) readily available or straightforward to derive? A brief discussion on the scope of manifolds where RCM is practically applicable would be insightful.
- The experiments focus almost exclusively on a 2-step generation scenario. How does the model's performance and the quality of the generated samples evolve as the number of inference steps increases to 4, 8, or 16? An analysis of this quality-vs-computation trade-off is crucial for understanding the model's behavior.
- Could you provide an analysis of the runtime complexity introduced by the covariant derivative calculation during training? How does this computational overhead compare to the Euclidean consistency models, and how is it expected to scale with the dimensionality of the manifold?

**Ethical Concerns:**

["NO or VERY MINOR ethics concerns only"]

**Final Justification:**

This work is very solid - I had some initial doubts, but after the rebuttal in which my main issues were addressed, I recommend accepting this work.

**Limitations:**

Yes.

**Paper Formatting Concerns:**

No.

**Quality:**

3

**Strengths And Weaknesses:**

## **Strengths**

- As far as I am aware, this work appears to be the first to develop a comprehensive and theoretically sound framework for consistency models on general Riemannian manifolds. This is a significant step forward, as it enables fast, high-quality generative modeling for a wide range of important scientific data (e.g., protein structures, robotics, geoscience) that naturally reside on non-Euclidean domains.
- The paper is technically very strong; both the the continuous-time RCM objective which seems a highly non-trivial result, but also the equivalence between the distillation (RCD) and training (RCT) variants  to further solidify the theoretical foundation. Also the kinematic perspective is really great, and the author are able to present rather intricate formalisms in a physical-intuitive manner making the paper quite accessible.
-  The authors demonstrate strong practical utility by not only presenting the full, complex objective but also proposing a simplified version (sRCM). This simplified loss eliminates the need to compute the differentials of the exponential map, which can be a significant implementation issue, yet it achieves excellent empirical results.


## **Weaknesses**
- The appendix shows that the method requires non-standard techniques like "Tangent Warmup" and "Tangent Clipping" to ensure stable training. The need for these specialized heuristics suggests that the proposed loss functions, while theoretically derived, may be inherently difficult to optimize and sensitive to the magnitude of the geometric correction terms. Could the authors reflect on how hard it is to work with the method in practice?
-  The practical application of RCM is challenging. First, applying the model to a new manifold requires deriving the specific forms of the exponential map, logarithm map, and Christoffel symbols for the covariant derivative. Can the author illustrate more for what manifolds this is doable?
- The empirical evaluation, while covering several manifolds, is limited in two key ways. First, the experiments are almost exclusively focused on a 2-step generation scenario, with no analysis of the quality vs. computation trade-off at other few-step counts (e.g., 4, 8, 16 steps), which would give significant insight in the general trend in comparison. Second, the datasets used are of a relatively small scale and low dimensionality. This makes it difficult to assess the method's scalability and performance characteristics on larger, more complex problems.
- The paper seems to not analyze the runtime complexity of the proposed method. The calculation of the covariant derivative term at each training step is computationally more expensive than the simple time derivative used in Euclidean models. This introduces an additional, unquantified computational overhead that may become significant for high-dimensional manifolds.

---

> ### Author Rebuttal · Authors · 2025-07-30
>
> We sincerely appreciate your recognition of the solid theoretical contributions and innovations in our work, which extend the existing consistency models on Euclidean data. We will address your questions as follows.
>
> ## Q1. Tangent Warmup
>
> We thank your insightful suggestions on the stability of the training stage of consistency models, and we are more than happy to share our findings on the practical training stage of our proposed RCM model.
>
> We first note that training stability has been recognized as a common challenge in many continuous-time consistency models, and not particularly for Riemannian manifolds. For example, the original CM paper did not run continuous-time CM on image generation, although it was also proposed; the ECT paper addressed such "curse of consistency" by gradually annealing the $\Delta t$; the sCM paper proposed many essential techniques for scaling up continuous-time CM models.
>
> In fact, the special techniques (e.g., tangent warmup and clipping) were directly inspired by the Euclidean sCM paper to address the issue that the initial JVP of the model, even when loaded from pre-trained weights, is noisy for the consistency constraint. Empirically, we did not find the additional geometric correction terms add special challenges to the training, and therefore, the standard tangent warmup technique is effective enough to stabilize the training process, allowing us to obtain decent and stable results across all continuous-time variants. As a concrete example on the Earthquake dataset with RCD, RCD was able to achieve 2.22 with both techniques, while no-warmup achieved 3.81, no-clip achieved 2.11, and no-warmup-no-clip achieved 2.70. It can be seen that the tangent warmup step is more important for the performance. While clipping has some negative impact on the performance, we empirically found it helps prevent gradient exploding in the early stage.
>
> ## Q2 Practical Applications to Different Manifolds
>
> We have proposed an alternative **simplified loss** in Section 3.2 that **only requires the additional information of the covariant derivative** on the manifold. The simplified loss eliminates the need for potentially complex symbolic calculations of the differentials of the exponential map for each different Riemannian manifold, achieving almost the same performance. In this way, the RCM framework can be easily applied to a wide range of manifolds. For example, in addition to the manifold we have experimented with, the standard hyperbolic manifold has known Christoffel symbol expressions, and all matrix Lie groups (e.g., SE(3), SU(2)) share a similar covariant derivative expression [1]. We will follow your suggestion to add a discussion in our revised manuscript to inspire potentially new applications.
>
> [1] Guigui, Nicolas, and Xavier Pennec. "A reduced parallel transport equation on Lie groups with a left-invariant metric." International Conference on Geometric Science of Information. Cham: Springer International Publishing, 2021.
>
> ## Q3 Quality vs NFE
>
> We provide additional ablation results on the model performance vs NFEs. Please kindly refer to our rebuttal to Reviewer aiD6, Q3. To summarize, in contrast to RFM, whose performance drastically drops when decreasing the NFEs, our RCM variants exhibit relatively stable performance. Such results demonstrate RCM can indeed shortcut the probability path on the Riemannian manifold. Our NFE vs performance results generally echo the findings in the consistency trajectory model [2], in which high NFEs may instead lead to poorer generations.
>
> [2] Kim, Dongjun, et al. "Consistency trajectory models: Learning probability flow ode trajectory of diffusion." arXiv preprint arXiv:2310.02279 (2023).
>
> ## Q4 Scalability
>
> Scalability has always been a key target for RCM. However, due to the limited availability of large-scale datasets in this domain compared to traditional image generation tasks in previous CM papers, we follow previous Riemannian generative models (e.g., RFM, Riemannian diffusion, Riemannian score matching) to focus on a wide range of different manifolds instead. Additionally, to demonstrate RCM's scalability on higher-dimensional manifolds, we have conducted experiments with a manifold dimension up to 256 following the RFM paper. Please refer to Reviewer aiD6, Q1, for the detailed results. To summarize, RCM **consistently outperforms RFM in few-step generation** even on higher-dimensional manifolds.
>
> In the runtime complexity analysis below, we further demonstrate that the overhead from the additional geometric components is both theoretically and practically negligible. Therefore, we believe RCM can be easily adapted for larger data and models, which we will continue to explore in future work.
>
>
> ## Q5 Runtime Complexity Analysis
>
> We provide a comprehensive analysis of the Riemannian operations. In short, we demonstrate **both asymptotically and practically** that the overheads of the additional Riemannian operations (e.g., covariative derivative, differentials of the exponential map) are **negligible**.
>
> Asymptotically, as the Riemannian operations used in the RCM algorithm are linear operators, the time complexity is $O(LD^2)$ where $L$ is the data dimension and $D$ is the number of manifold dimensions where the Jacobian is non-diagonal. (Therefore, the n-torus is actually the data dimension of n because it is the direct product of n 1-tori.) In practice, $L$ can be large (e.g., number of pixels in the Euclidean case), but $D$ is limited, so the behavior **scales linearly as the data dimension and is model-independent**. Practically, we observed that the model's forward and backward passes were the bottleneck. Specifically, the relative overheads during training are provided below.
>
> | overhead/% | RCM  | sRCM |
> | ---------- | :--: | :--: |
> | Earth      | 5.9  | 4.1  |
> | RNA        | <0.1 |  -   |
> | SO(3)      | 10.7 | 7.8  |
>
> It can be clearly seen that, even with a fairly small model, the additional Riemannian operations have little overhead in general. It is also expected that the overhead for SO(3) is larger than the sphere, as the former requires more operations, like the calculation of left/right Jacobians. However, it is still within 10% and will be even smaller for larger models.

---

> > ### Comment · Reviewer_nYt4 · 2025-08-06
> >
> > Dear authors,
> > Thanks for the extensive rebuttal. You addressed my points well and as such I raised my score.

---

> > > ### Author Response · Authors · 2025-08-06
> > >
> > > We sincerely appreciate your insightful review and positive feedback, and we are delighted that our clarifications and additional results have successfully addressed your questions. We are committed to further polishing our RCM and will include the additional results in our revised manuscript to make our work more comprehensive and solid.

---

### Official Review · Reviewer_Ezj8 · 2025-07-07

**Clarity:** 3
**Significance:** 3
**Originality:** 3
**Rating:** 5
**Confidence:** 3

**Summary:**

This paper presents an extension of Consistency Models to the case of non-Euclidean Riemannian manifolds. The ability to accelerate diffusion/flow matching on such manifolds is impactful for a number of scientific applications ranging from protein generation to astrophysics and had until this paper not received sufficient attention. The paper develops the key results necessary to derive both consistency distillation and consistency training on Riemannian manifolds, introduces simplified objectives that bypass some of complexities of geometric calculus, and performs adequate evaluations of the different variants of the proposed models on established and appropriate benchmark problems.

**Questions:**

- I noticed a missing stop gradient in Eq 2

**Ethical Concerns:**

["NO or VERY MINOR ethics concerns only"]

**Final Justification:**

This is a sound paper that develops the required results for building consistency models on Riemannian manifolds and I am satisfied with the practical evaluations of the proposed method.

**Limitations:**

Yes

**Quality:**

3

**Strengths And Weaknesses:**

**Strengths**:
- The paper is very well written, provides appropriate background and survey of the related literature, and includes useful insights to help the reader understand the different aspects of the consistency loss on Riemanian manifolds.
- The derivations are mathematically sound (as far as I can tell), providing the necessary key results to build consistency models on Riemannian manifolds.
- The evaluations are appropriate and demonstrate 1. the gain of using a proper riemannian distillation strategy over a naive implementation of a Euclidean consistency model in the ambient Euclidean space, 2. the speed up made possible by consistency modeling compared to standard flow matching.

This is a well rounded paper, I do not have significant weaknesses to point out.

---

> ### Author Rebuttal · Authors · 2025-07-30
>
> We sincerely appreciate your high recognition of our work's clear structure, solid mathematical background, and comprehensive evaluation. As per request by the other reviewers, we also provide the following additional experiments to make our work more comprehensive.
> - Scalability to higher-dimension manifolds with a dimension up to 256, in Reviewer aiD6, Q1. RCM consistently achieves better generation quality.
> - NFEs vs performance evaluation, in Reviewer aiD6, Q3. RCM has a stable performance compared to RFM.
> - Additional FFF baseline results (a Riemannian normalizing flow), in Reviewer J7YW, Q6. RCM consistently outperforms FFF.
>
> We will continue to refine our work and explore the important applications of our proposed RCM in various AI4Science domains, where Riemannian manifolds are frequently encountered. We hope that our framework, as one of the earliest attempts to introduce a mathematically sound approach to Riemannian few-step generation, can also inspire future work.

---

### Note · Authors · 2025-08-14

We sincerely thank all the reviewers for their time and effort throughout the review process and their insightful reviews and feedback that helped make our work more comprehensive.
We are also glad that our clarifications and additional experimental results have **successfully addressed the questions and concerns** in the original reviews, and our **theoretical contributions and technical significance** of RCM in the few-step generative modeling setup on Riemannian manifolds have been unanimously acknowledged by all reviewers.
We thank Reviewer nYt4, aiD6, iNgD, and J7YW for raising their scores, and we are also grateful for Reviewer Ezj8's consistent high recognition and support of our work.

We appreciate all the reviewers for their insightful suggestions on additional experiments and ablations. For example, Reviewer nYt4, aiD6, and J7YW suggested applications to higher-dimensional manifolds to demonstrate the scalability of RCM; Reviewer nYt4 and aiD6 suggested an investigation into the impact of NFEs; Reviewer J7YW also mentioned FFF as an additional baseline. We have carried out these additional experiments and were encouraged to observe the **consistently superior performance of RCM variants over the naive Euclidean baseline**. We also appreciate and enjoy the insightful discussions with all the reviewers, e.g., the NLL discussion with Reviewer J7YW, which may lead to a potential future extension of RCM. We are committed to further polishing our RCM and will include the new results in the revised manuscript for a better few-step Riemannian generative model.

---

### Decision · Program_Chairs · 2025-09-17

**Decision:**

Accept (poster)

**Comment:**

This paper proposes to extend consistency models to Riemannian manifolds. All reviewers agree that the method is technically sound and leads to a new class of few-step generative models for geometric domains. Despite this, the reviewers and I agree that the work in terms of novelty is incremental as it combines known ideas in consistency models and Riemannian generative models in a straightforward---but well executed---manner. This means that the theoretical results are used to substantiate the algorithm design, but by themselves don't offer profound implications on Riemannian generative models. In addition, the experimental evaluation is limited and higher-dimensional datasets are not considered (e.g. protein backbone generation)---despite the authors' implication in the rebuttal. Having said all of this, I still believe this paper fills a gap in the literature and is an appropriate contribution to the community to deserve publication at NeurIPS.